

# Linking Alpine deformation in the Aar Massif basement and its cover units – the case of the Jungfrau-Eiger Mountains (Central Alps, Switzerland)

David Mair[1], Alessandro Lechmann[1], Marco Herwegh[1], Lukas Nibourel[1], Fritz Schlunegger[1]

[1] Institute of Geological Sciences, University of Bern, Baltzerstrasse 1+3, CH-3012 Bern

*Correspondence to*: David Mair (david.mair@geo.unibe.ch)

**Abstract.** The NW rim of the external Aar Massif was exhumed from ~10 km depth to its present position at 4 km elevation above sea level during several Alpine deformation stages. Different models have been proposed for the timing and nature of these stages. Recently proposed exhumation models for the central, internal Aar Massif differ from the ones established in the covering Helvetic sedimentary units. By updating pre-existing maps and collecting structural data, a structural map and tectonic section was reconstructed. Those were interpreted together with micro-structural data and peak metamorphic temperature estimates from collected samples to establish a framework suitable for both basement and cover. Temperatures at deformation ranged from 250°C to 330°C allowing for semi-brittle deformation in the basement rocks, while the calcite dominated sediments deform ductile at these conditions. Although field data allows to distinguish multiple deformation stages before and during the Aar Massifs rise, all related structures formed under similar P, T conditions at the investigated NW rim. We find that the exhumation occurred during 2 stages of shearing in the Aar Massif basement, which induced in the sediments first a phase of folding and then a period of thrusting, accompanied by the formation of a new foliation. We can link this uplift and exhumation history to recently published large-scale block extrusion models.

## 1 Introduction

The Aar Massif is the largest External Crystalline Massif (ECM) in the Alps, made up of exhumed pre-Triassic basement rocks and Mesozoic to Cenozoic cover sediments along its NW to SE striking frontal margin. In this region, the Eiger, Mönch and Jungfrau mountains in the Swiss Alps feature an immense topographic expression along this SE-NW-striking rim, with almost 1800 m of vertical offset in their north faces. Throughout their stepwise, pyramidal headwalls these mountain ranges expose both the crystalline substratum and the sedimentary cover rocks of the Aar Massif. These scenic outcrops are thus key to understanding the Massif's exhumation from ~10 km depth to its present position at 4 km elevation above sea level. Therefore, these mountains have been the focus of a long tradition of structural research, which yielded a general picture of a steeply dipping autochthonous cover sediments in front of an up-domed ECM (i.e. Pfiffner, 2014). Further to the NW the detached fold-and-thrust nappes of the Upper Helvetic are considered to have been decoupled from the massif's evolution in an early stage, and the Helvetic units have experienced a passive up-doming in response to the rise of the Aar Massif after their displacement into a frontal position in the NW during the Oligocene (i.e. Schmid et al., 2004,





Hänni & Pfiffner, 2001). The lithostratigraphic and tectonic studies, which resulted in the reconstruction of this scenario, have been conducted over the course of more than 150 years and have been focused on a few key regions of this ECM. These mainly include: The S and SW sectors of the Aar Massif (i.e. Krayenbuhl & Steck, 2009; Herwegh & Pfiffner, 2005; Steck, 1984; Steck, 1968), and the area surrounding the Jungfrau and the Mönch mountains (i.e. Rohr, 1926; Scabell, 1926;

Collet and Paréjas, 1931; Günzler-Seifert & Wyss 1938; Kammer, 1989). Farther to the NW in the region of the Bernese Oberland, the neighboring Mesozoic sediments have been studied later in detail (i.e. Hänni & Pfiffner, 2001; Menkveld, 1995; Pfiffner, 1993 and references therein).

The best-studied region of the Aar Massif is the Haslital, which stretches from Innertkirchen up to the Grimsel Pass (Abrecht, 1994 and references therein) and which exposes the crystalline rocks of what has been referred to as the Central

Aar Massif. A swath of petrological and chronological data (i.e. Challandes et al., 2008; Rolland et al., 2009; Schaltegger et al.; 2003) together with structural observations (i.e. Wehrens et al., 2017, Wehrens et al., 2016, Berger et al., 2017a) advanced our understanding on the geodynamic evolution of the Aar Massif. This culminated in a new model for the exhumation of the Aar Massif (Herwegh et al. 2017) and in a new regional-scale geological map (Berger et al. 2017b). Nevertheless, details about how the tectonic deformation affected the crystalline basement and the sedimentary cover rocks

of Aar Massif, and if and how this deformation propagated into the sheared-off Helvetic nappe system in front of the Massif have not been explored in detail for the Jungfrau-Mönch area. This is mainly due to the complexity of the geologic architecture and the inaccessibility of the area that have thwarted a precise reconstruction of the history and the amount of shortening of the frontal part of this Massif. It is the scope of this paper to link the tectonic history of these uplifted basement blocks to the structures in the cover and to fill this knowledge gap.

Here, we reconstruct the relative chronology of the frontal part of the Aar Massif in 3D. We focus on the central part presently exposed in the Central Swiss Alps, where this contact is exposed c. 12km along the strike of the basement cover boundary. We proceed through (i) establishing a synthesized stratigraphic framework for the region, (ii) collecting new structural data and samples in the field and along the "Jungfraubahnen" railway tunnel that crosses the mountain range, and through (iii) modeling the tectonic architecture with GIS and Midland's Move™ software package. We differentiate the

cover sediments based on stratigraphic criteria, which in turn allows us to reconstruct the geometry of the exposed units. In addition, structural data analysis enables us to unravel their relative deformation history, while Raman Spectroscopy on Carbonaceous Material (RSCM) yields estimates on peak metamorphic temperature. Our synthesis of existing data together with new observations finally allows us to link the fabric of the cover sediments with the underlying basement units' evolution for one of the Alps' most famous scenery.



## 2 Geological setting

### 2.1 Tectonic architecture

The Aar Massif is the largest External Central Massif (ECM) in the Alps and is made up of polymetamorphic pre-Variscan gneisses with intruded post-Variscan granitioids (Labhart, 1977; Abrecht, 1994). The most external polycyclic gneiss unit is exposed along the Aar Massif's northwestern rim, referred to as the Innertkirchen-Lauterbrunnen zone (ILZ; Berger et al., 2017a; Abrecht, 1994; Fig. 2). Farther to the SE, the Erstfeld Zone exposes gneiss units (EZ; Abrecht, 1994) and occurs in a hanging wall position to the ILZ (Berger et al., 2017b; Fig. 2) with sediments squeezed in-between. This tectonic sliver made up of sedimentary and crystalline rocks is referred to as Jungfrau-Sediment-Wedge (JSW). Both units share a concordant overall SW-NE strike direction of structures such as lithological boundaries and foliations (Oberhänsli et al., 1988). These pre-Variscan basement units experienced multiple periods of deformation and metamorphic overprint, which occurred during the Proterozoic, Ordovician, Variscan, the Late Cretaceous and the Cenozoic (Steck, 1968; Labhart, 1977; Schaltegger, 1993; Schaltegger et al., 2003). In our study area (Fig. 1) and farther to the west, the EZ is separated from the ILZ by this wedge of Mesozoic sediments (Krayenbuhl & Steck, 2009; Herwegh & Pfiffner, 2005; Steck, 1968). Additional autochthonous Mesozoic cover sediments are present at the NW rim of the Aar Massif (Kammer, 1989), where they form an own, detached and transported nappe system (Doldenhorn nappe; Herwegh & Pfiffner, 2005).

The sediment wedges and the Mesozoic cover were only affected by Alpine deformation and metamorphism. In the study area, the Alpine metamorphic overprint occurred under lower greenschist metamorphic conditions (Frey & Mählmann, 1999; Niggli & Niggli, 1965). The peak temperatures increased towards the SE, where conditions of ~450°C and 6.5 kbar have been reconstructed for the Cental Aar Massif granitoid shear zones (Challandes et al., 2008) at a time around 20 Ma (Wehrens et al., 2017; Herwegh et al., 2017).

### 2.2 Alpine structural evolution

The structural imprint of this ECM has been related to various deformation stages by multiple authors (Table 1), often depending on the site-specific conditions. This resulted in the generally accepted notion that during the late Eocene, the Helvetic sedimentary nappes were detached from their crystalline substratum situated farther to the SW (Pfiffner, 2014; Herwegh & Pfiffner, 2005; Burkhard, 1988). These mechanisms are considered not to be recorded by the structural fabric in the Aar Massifs basement (Wehrens et al., 2017; Berger et al., 2017a). This has been used as argument in geodynamic work to disconnect the evolution of the basement rocks from that of the cover sediments. An early deformation fabric assigned to the "Plaine Morte" phase of deformation is recorded in the western Central Helvetic units only (Burkhard, 1988; Pfiffner, 2014). The subsequent Oligocene phases of deformation, which were referred to the "Prabé" phase in the west and "Calanda" phase in the east of the Helvetic nappes (Burkhard, 1988; Pfiffner, 2014), were associated with the main phase of thrusting within the Helvetic units and are recorded by a penetrative foliation. Further shortening led to the formation of the





Doldenhorn nappe (former Infrahelvetic and new Lower Helvetic), when a former half graben basin was inverted and incorporated into the Alpine edifice. This phase of deformation, which has been referred to as the "Kiental" phase (Herwegh & Pfiffner, 2005; Burkhard, 1988), produced a large-scale recumbent fold and likewise induced passive folding in the overlying Helvetic nappes. By the end of this phase, around 20 Ma, the Helvetic nappes were placed in front and on top of

the future Aar Massif with an inverse succession, where the Autochthon and the Doldenhorn nappe were below these nappes (Herwegh & Pfiffner, 2005).

The exhumation history of the crystalline basement rocks of the Aar Massif, generalized as "Grindelwald" phase (Güntzler-Seifert, 1945; Burkhard, 1988), records the following multistage late-Alpine deformation sequence: (i) First, steeply south dipping reverse and normal faults developed a set of pervasive shear zones during the "Handegg" phase of deformation

(Wehrens et al., 2016; Wehrens et al., 2017), with a progressively increasing differential uplift component towards the south (Herwegh et al., 2017). Subsequently, a phase of strain partitioning occurred simultaneously with (ii) dextral strike slip to oblique slip shearing along NW-SE and WNW-WSE trending faults ("Oberaar" phase) in the south and (iii) NW directed thrusting ("Pfaffenchopf" phase) along moderately southeast dipping fault planes (Labhart, 1966; Wehrens et al., 2016; Wehrens et al. 2017; Herwegh et al., 2017; Berger et al., 2017a). During this latter deformation phase, the uplift passively

rotated the former main thrust faults and foliation of the Helvetic and Doldenhorn nappes (Burkhard, 1988). This supposedly lead to the present, almost vertical orientation of the main Helvetic thrust in front of the Eiger (Pfiffner, 2014). The latest deformation stage ("Gadmen" phase) is recorded by a steep, NE-SW trending northern block with brittle deformation structures (Labhart, 1966; Berger et al., 2017a).

## 3 Methods and data

A regional structural map was produced (Fig. 2) with the aid of remote sensing techniques. This map was reconstructed by compiling geological and structural information from previous maps (for a review of the data sets used see Appendix A). These were verified and updated by own mapping in the field. Special focus was directed towards the mapping of stratigraphic contacts in the cover sediments and shear zones in the basement, both on outcrop and regional scales. Field mapping was done using the 1:25'000 Topographic Map of Switzerland as basis, which was enlarged to a working scale of

1:10'000. We additionally employed high resolution orthophotos (raster resolution 0.25 x 0.25 m; provided by swisstopo) and a high-resolution digital elevation model (swiss ALTI3D, version 2013 provided by swisstopo) as bases for mapping. Structural data (orientation of bedding, lithological contacts, foliation, lineation and faults) were collected with a traditional geological compass and a handheld GPS. The structural dataset was expanded by producing a lineation map thereby following the workflow of Baumberger (2015), Baumberger et al. (in press) and Schneeberger et al. (2017), which in turn is

based on Rahiman & Pettinga (2008). We mapped only lineaments that were visible both on the DEM and aerial images, and that were readily identifiable in the field. Lineaments that were observed on remotely sensed datasets only are indicated as

"inferred". Orientations thereof were obtained by plane fitting through moment of inertia analysis of georeferenced point data using the method of Fernández (2005). In addition, we completed the geologic map through collection of geologic information in the "Jungfraubahnen" railway tunnel. The tunnel runs oblique to parallel to the striking direction of the main
structural elements between ~3000 m elevation in the E and ~3400 m in the west (Fig. 2).

The map was combined with microstructural observations on thin sections. Thin sections were cut parallel to stretching lineation and normal to foliation planes, thus allowing shear sense directions to be identified. In addition, we used thin section observations to qualitatively estimate the temperature conditions during peak metamorphic conditions and during periods of dynamic recrystallization of quartz and calcite aggregates.

Raman spectroscopy on carbonaceous material (RSCM) was used for peak temperature estimations recorded by the Mesozoic sediments. The RSCM technique quantifies the degree of graphitization in meta-sediments, which is a reliable indicator of peak metamorphic temperature (Beyssac et al., 2002). RSCM measurements were performed with a Jobin Yvon LabRAM-HR800 instrument at the Institute for Geological Sciences at the University of Bern. An Nd-YAG continuous-wave laser (20 mW beam spot of 1 μm diameter and wavelength of 532.12 nm) focused through an Olympus BX41 100x
confocal microscope was used. The acquisition of the Raman spectra was monitored with the Labspec 4.14 software of Jobin Yvon. Curve fitting (for histograms see Appendix B) and temperature estimation followed strictly the procedure described in Lünsdorf et al. (2014), Lünsdorf and Lünsdorf (2016) and Lünsdorf et al. (2017). The absolute temperature calibration-based error is in the order of ±40°C, however, relative temperature differences can be resolved down to ±15°C (Lünsdorf et al., 2017).

# 4 Results and interpretation

We present an inventory of structural rock fabrics (both on outcrop and microscale) of the studied area. These fabrics developed during Alpine deformation and are different for the cover sediments and for the basement rocks.

## 4.1 Host rock characteristics

### 4.1.1 Polymetamorphic Aar Massif basement rocks (ILZ & EZ)

The basements rocks of the ILZ and the EZ are present as plagioclase, alkali feldspar and quartz dominated gneisses, which are occasionally enriched in micas (mainly biotite and white micas) and chlorite minerals (Fig. 3, second row). In the study area, the texture ranges from typical granoblastic (the largest grain size of feldspars is < 2 cm) to granodioritic (grain sizes of <<1mm) for the matrix zones of the migmatites (Rutishauser, 1973). Usually the feldspars (both alkali feldspar and plagioclase), together with biotite, form large grains with interstitial quartz. This texture is only partially preserved, due to a
later greenschist facies overprint.

This overprint is recorded by chlorite replacing biotite and quartz, sericitization of feldspar grains, as well as by the growth of interstitial white mica (see also Berger et al. 2017b). The overprint, however, did not erase the original high temperature fabric completely, instead it often forms core-rim structures around altered feldspar grains (Fig. 3: second row). Quartz aggregates still preserve the original texture. No new biotite growth occurred and no preferred orientation (neither for quartz

nor micas) on the microscale is found. This aligns well with the lack of evidenced for an Alpine foliation on the outcrop scale in some basement rock outcrops outside discrete shear zones.

### 4.1.2 Mesozoic cover sediments

The Mesozoic cover sediments form an originally up to 500m-thick succession of limestones, mudstones and sandstones (Fig. 4; for detailed discussion see Appendix A). The stratigraphic suite can be synthesized into eight larger units (Fig. 4 &

Table 2). This allows to distinguish the main stratigraphic horizons, and to group units with similar mechanical strengths (see also Pfiffner, 1993; Sala et al., 2014). These units comprise: the Triassic (Mels-, Quarten- and Röti Fm.), the Mid Jurassic (Bommerstein- and Reischiben Fm.; "Dogger"), the Upper Jurassic A (Schilt Fm.) and B (Quinten Fm.), the Lower Cretaceous A (Öhrli Fm.) and B ("Helvetic Kieselkalk" and Betlis Fm.), the "Siderolithic" and the Tertiary units.

There are several main characteristics needed to understand the structural fabrics within the individual sedimentary units.

The basal Triassic, above the weathered basement-cover-contact, is formed by crystallized cellular dolomites and shales (Appendix Fig. A1). Despite varying thicknesses (5m- 50m) and stratigraphic contents, it is present throughout the entire study area. The 250m to 300m-thick suite of Upper-Jurassic limestones, and the overlying up to 150m-thick succession of Lower Cretaceous limestones form the bulk of the strata and are constant in thickness throughout the studied area. The only difference between the facial domains (Fig. 4; aside from a thicker Lower and Mid Jurassic strata) is the presence of a

Cretaceous B unit (in the northern flanks of Jungfrau and Eiger), which consists of layered cherts with limestone interbeds. Note that these features allow its identification as continuation of the Doldenhorn nappe farther to the west.

### 4.2 Deformation structures

The studied mountain chain is dissected by numerous high strain zones with variable orientations, which occur both in the basement and in the sedimentary cover. Note that we refer to as shear zones for fabrics, which formed through ductile or

semi-ductile deformation mechanisms. Only for brittle dominated structures (cataclasite and gauge dominated) we apply the term fault.

### 4.2.1 Basement strain localization

Locally, a weakly pronounced, pre-Alpine foliation can be found. If preserved, it is usually cut or overprinted by Alpine exhumation related structures. These latter structures generally occur as discrete sub-mm- to meter-thick shear zones in the

field (Fig. 5a).  They are present either as a set of generally steep or as a set of intermediate S dipping semi-brittle to brittle





shear zones (Fig. 5). The steep shear zones, which we define as SZ2 (Figs. 3 & 6; Note that SZ1 shear zones were only found in the sediments), exhibit normal and reverse faulting behavior with dominantly top to the NW shear senses, as is indicated by the offset of older structures (Fig. 5b). However, several individual faults show evidence for reverse movements as documented by displacements of isolated dm-sized blocks. Typical Riedel shears are present, highlighting the conjugate

semi-brittle nature of the shear zones. The spacing between them decreases towards the NW and close to the JSW. In the Rottal (RT), the mean orientation of these structures is 315/84 (dip azimuth/dip) with south block up movement. In the Trugberg (TB) zone the average dip azimuth is 180/52 (inferred from remote sensing). Note that owing to younger deformation overprint, the original orientation might have been different than at present.

The intermediate steep set of shear zones (SZ3; Fig. 6) cuts and offsets the SZ2. structures. The shear zone spacing is much

wider and the deformation occurred more localized (especially within the JSW). The orientation of the SZ3 suites also changes from moderately steep SE dipping to NW dipping in the NW (TB: 157/22 vs. RT: 351/27) with NW plunging stretching lineations (RT: 322/21).

Despite the differences in orientation and kinematics, the microfabrics of both SZ2 and SZ3 samples show no difference in terms of deformation microstructures and mineral assemblages. Grains of the original pre-Alpine granoblastic fabric are

either replaced by new grains of much smaller size (i.e. chlorite, white mica, epidote; Fig. 3; 3rd row), which are well-foliated, or they experienced mechanical grain size reduction down to a few micrometers. Those small-grained minerals are often concentrated in micrometer to centimeter thick polymineralic bands. They form SC fabrics and shear bands (Fig. 3). However, feldspar and quartz minerals are still present as porphyroclasts despite evidence for incipient dynamic bulging recrystallization for quartz (see also Bambauer et al., 2009). Notable is the higher phyllosilicate content in shear zone rocks

compared to the host rock composition.

A youngest set of (i) NW-SE running, sub-vertical fault planes ("ac-joints" of Ustazeswksi et al., 2007) and (ii) SE-NW striking steep faults (Fig. 6c and Fig. 9b) that cut all older structures and are characterized by brittle deformation forming cataclasites, fault breccias and gauges (Appendix Fig. A2). Both fault sets cross-cut all other structures and are best detected in the subsurface (along the "Jungfraubahnen" tunnel; Fig. 2), due to their susceptibility to weathering. Set (i) shows oblique

to strike-slip kinematics (Fig. S3) in cm to m wide cataclasitic shear zones, often with fault gauges at the core. Set (ii) mainly features mm to cm wide shear zones along a set of open joints. However, the offset of these faults is in the order of a few centimeters to meters and is thus not resolvable in Figures 2 and 10.

### 4.2.2 Deformation in the cover sediments

The cover sediments show strong evidence for ductile deformation and brittle deformation, both on outcrop scale and in thin

sections. Mapping reveals that the deformation fabric is dependent on the rheology of the hosting rocks. The calcite-dominated limestones exhibit a strong ductile overprint, expressed by complete dynamic recrystallization of the original fabric with grains that are smaller than 25 µm (Fig. 8). They show a well-developed foliation (S1) parallel to the bedding,



which formed through abundant calcite dissolution and dynamic recrystallization. This first mylonitic foliation shows a spacing ranging from several cm in the lower RT (Rottal) section to <<1mm in the Mönchsjoch (MJ). Typically, it overprints

the bedding completely (Fig. 7). Notably the Triassic dolomite (Röti Fm.) was not deformed in a ductile, but rather in a brittle or semi-brittle manner. This is expressed in abundant bookshelf structures and "domino-like" boudins affected by more than one deformation phase (Fig. 7c). These rigid lenses are stretched out along shear planes or form kink folds near the basement cover interface. The same applies to mid Jurassic iron-rich sandstones and Cenozoic iron-rich sandstones or iron-carbonate nodules (Siderolithic). This led to macroscopic (and microscopic)-scale boudins (Figs. 7 & 8) making these

units ideal stratigraphic markers to identify shear zones and stratigraphic polarity.

In the NW flank of the Jungfrau-Eiger mountains, the Mesozoic sediments occur in distinct stacks, separated by arrays of discrete shear zones (SZ1) of < 1 m to up to 5 m widths that formed during the formation of S1. These shear zones were accompanied by the circulation of metamorphic fluids, as testified by abundant iron-rich, micrometer-thick layers of precipitated minerals. Growth of white mica and chlorite minerals within the pressure shadows is frequently observed,

mainly in non-calcite dominated lithologies. These thrusts contain slivers of ILZ basement rocks in the footwall, which are now incorporated in the sedimentary stacks (i.e. in the Rottal area, Fig. 2).

Subsequent anti-clockwise rotation of the initial structures (bedding and S1, in section view, looking E) is documented by rotated sigma clasts in the shear zones (and the mineralized veins within them; Fig. 7). As a result, the present-day orientation of S1 varies form SE dipping (118/35 for EN, 118/36 for EW) in the lower para-autochthonous slice, to the NW

dipping (312/22 for GG; 297/36 for EM; Fig. 9). The original orientation is not preserved. This deformation stage affected sediments in an asymmetric way, and the folding is intensified towards the SE of the studied area. Consequently, these structures culminate in a weak dm-spaced axial plane foliation (S2) in the JSW.

Subsequent NW-directed shearing led to a further thinning of favorably SE dipping or flat lying aligned thrust-planes of S1 fabrics in the now flat lying limbs of the stacked imbricates (Fig. 11b). The new foliation (S3) formed parallel (or only

cutting under low angles) to the pre-existing mylonitic foliation. In case where S1 structures dip in an unfavorably steep NW orientation (as it is the case in RT region or at the location of Fig. 5), the local new foliation (S3) developed still under the same orientation, cutting the older foliation (S1) at high angles. This foliation features evidence for slip where S1 fabrics are cut. Dip azimuth alters between gently SE dipping, flat lying and NW dipping, with a dominant top to NW thrust, indicated by stretching lineation data derived from abundant calcite slickenside striations on S3 slip planes. This shearing occurred still

under ductile conditions for calcite, however non-limestone dominated lithologies (including incorporated basement slivers) exhibit shear localization merely by mechanical grain size reduction of minerals to a few microns. They form a microcrystalline cataclasite consisting mainly of quartz, white mica, chlorite, epidote and feldspar.

The latest deformation stage produced a set of steep SW and NE striking oblique brittle fault zones (SZ4) and brittle sub-vertical faults striking NW-SE. Both are found within sediments and cover rocks of all levels alike (documented by fault

breccia and mineralized veins crosscutting an older set; Appendix figure A2). These overprint all younger structures. Due to



the brittle nature and connection to the surface they are often water saturated and highly susceptible to weathering. Despite cutting across all units, offsets (if even present) are limited to a few centimeters only and are thus not visible on map scale.

### 4.3 Peak metamorphic temperature

Raman spectroscopy applied to carbonaceous material (RSCM) was used on 8 selected samples representing the different tectonic levels in the studied area to constrain peak Alpine metamorphic temperatures for the cover sediments (Table 3). Results yield the lowest temperatures for the presently lowest elevation sample (LAU-02: 283±14°C at 838 m a.s.l. for the Upper Jurassic limestone) and samples from the Eiger north flank (EN-01: 283±12°C at 2388 m a.s.l.). Slightly higher peak temperatures are obtained for samples collected at the Eiger summit (EG-17-01: 292±10°C at 3970 m a.s.l.). Highest temperatures are found in the JSW samples (MJ-03 and MJ-06: 308±14°C and 317±11°C respectively). The data indicate a

trend to slightly higher temperatures towards the more internal units in the SE. Assuming a constant geothermal gradient of 25°C km$^{-1}$ (Pollack & Chapman, 1977) for the past, we are able to reconstruct the sample depth at peak metamorphic conditions. This allows us to identify the vertical position of the corresponding units relative to each other. The overall temperature range of 283°C to 317°C indicates a sample depth between 11 and 13 km at peak temperature conditions. This further constrains the peak temperature for the crystalline substratum of these sediments to <330°C.

### 4.4 Imbricate geometry in the sediments

Stratigraphic markers (see Sect. 4.1.2) are used to delineate geometrical bodies separated by thrusts of different generations (see Sect. 4.2.2). From the NW to the SE (and present-day bottom to top) we find the ILZ and its sedimentary cover still in its original position. This is superimposed by two imbricate stacks of c. 500m thickness, with normal stratigraphic succession within each stack (often referred to as "para-authochthonous", Fig. 10). On top of these, we additionally find a >1000 m-

thick pile of Cretaceous and Jurassic limestones, which was often referred to as the core of a syncline (Krayenbuhl & Steck, 2009). These units are deformed as an ensemble during stage 2 and stage 3 of deformation as described in Sect. 4.2 (see Fig. 11). The subdivision of this sedimentary stack is consistent along strike within the study area. Lateral differences in structural style, which occurred in response to the last 2 deformation stages, are expressed by less shortening along the JSW, while more basement slabs were detached and thrusted at a lower level. The ensemble of this deformation pattern resulted in

a steeper (almost vertical) orientation of the basement cover contact.

### 5 Discussion

The cover sediment reveals a set of distinct deformation fabrics that formed during a 3-stage evolution. We derive these stages from aforementioned field data and geometrical relationships. By disentangling the fabrics related to each stage, we can link these in a regional geodynamic context.



### 5.1 Pre-Alpine inheritance

An important role falls to Paleozoic and older structures that are inherited in the basement (i.e. the polymetamorphic basement called "Altkristallin" in the German literature; Steck, 1968). Since the ILZ and EZ were originally sedimentary protoliths (Rutishauser, 1973; Rutishauser, 19734; Schaltegger, 1993) they feature a heterogeneous architecture (i.e. Abrecht, 1994). Large scale partial melting produced the original host rock fabric (Sect. 4.1.1). Radiometric dating yielded Ordovician ages for the of the Erstfeld zone (456±2 Ma) and the Innertkirchner Lauterbrunnen migmatites (452±5 Ma; Schaltegger, 1993) for this high temperature metamorphic overprint. Thus, any evidence of a previous geological history was erased, but the metasomatic overprint preserved to some extent the original heterogenous lithological character (Berger et al., 2017a, Abrecht, 1994). Subsequent tectonic events already aligned structures along a SW-NE direction, i.e. lithological boundaries and foliation (Schaltegger et al., 2003). The wedging of Permian volcanoclastic sediments, which was associated with folding, suggests that the basement internal nappe emplacement occurred already during the Carboniferous (Oberhänsli et al., 1988). The emplacement of several late to post-Variscan granitic intrusions completed the pre-Mesozoic evolution and presumably lead to the greenschist facies overprint in the host rocks. The resulting heterogeneities were intermediate to steep S to SE dipping fabrics (Berger et al., 2017a), which already formed before the initiation of the Alpine orogenic cycle and therefore represent important mechanical anisotropies for the succeeding tectonic evolution (Herwegh et al., 2017).

### 5.2 Stratigraphic priming

Some of the steep SE orientated pre-Alpine heterogeneities were reactivated during the Mesozoic as normal faults within the Helvetic shelf of the Tethys Ocean (i.e. Hänni & Pfiffner, 2001). Strikingly, in the study area the JSW sedimentary wedge (that should later act as a major thrust) is located at the pre-Alpine boundary between the ILZ and EZ. The favorable orientation most likely resulted in the re-activation as Jurassic normal faults, causing the stratigraphic NW-SE asymmetry in the Mid-Jurassic (Herwegh and Pfiffner, 2005; Krayenbuhl & Steck, 2009). This allows to account for the evolution across the former Helvetic shelf. Two important stratigraphic observations can be made in the Mesozoic of the Jungfrau tectonic sliver and associated shear zones: (i) The Triassic sequence is eroded to a deeper level towards paleo SE, owed to the asymmetric Liassic erosion ("Alemannic land": Pfiffner, 2014; Rohr, 1926) and (ii) the subsequent sedimentation in the Middle Jurassic is governed by normal faulting, resulting in thicker sediment successions at deeper water conditions to the SW. Apart from these differences (and the difference in the Lower Cretaceous) the post-rift cover sediments are rather similar in thickness and facies throughout the studied area.

Generally, the autochthonous cover of the Aar Massif comprises a Mesozoic stratum resembling the northwestern-most facies of the Central Helvetic domain in the Lauterbrunnen valley (Bruderer, 1924; Masson et al., 1980; Herb, 1983). On a larger scale, our stratigraphic model aligns well with recent findings, i.e. for the Triassic (Gisler et al., 2007) and for the revised Tertiary stratigraphy of the Helvetic realm (Menkveld-Gfeller et al., 2016). This stratigraphic model (see Fig. 4 and





table 2) allows to bracket unit thicknesses. We are aware of the partly large uncertainties on these values (sometimes up to 100%), yet it is still useful for omitting unrealistic geometries.

## 5.3 Structural imprint of the Alpine evolution

### 5.3.1 Early stage deformation under highest Alpine temperature conditions

Peak metamorphic conditions are constrained by the RSCM estimates to low temperature (sub-greenschist facies) conditions for the cover sediments and the immediate crystalline substratum. An upper limit for the calcite thermometry is provided by the graphite data from the most internal part of the JSW (Table 3) at ~320°C. Since the bulk of the Mesozoic strata consists of limestones (Fig. 2) the temperature range for the deformation has a lower constraint of 200-250°C, which governs the onset of the ductile deformation of calcite (Herwegh et al. 2005, Burkhard 1990). These conditions align well with reported

regional Alpine metamorphic gradients (i.e. Herwegh et. al., 2017; Niggli & Niggli, 1965). Only at temperatures as high as ~300°C or above, onset of ductile deformation in quartz occurs, as can be inferred from the occurrence of dynamic recrystallization in form of bulging recrystallization (e.g., Stipp et al., 2002; Bambauer et al., 2009; Härtel and Herwegh, 2014). However, our quartz-rich basement rocks were mainly deformed in a semi-brittle manner (see. Sect. 4.2.1) as Quartz and Feldspar are in general mechanically reduced in grain size within discrete shear zones (Fig. 3; last row). The mylonitic

character is primarily owed to ductile behavior of micas, chlorites and the fine-grained polymineralic gauges (similar to findings reported by Wehrens et al., 2016; and Wehrens et al., 2017). We see the onset of dynamic recrystallization in quartz through evidence for bulging in some samples. Thus, we must place the development of all shear zones in the basement close to or shortly after peak metamorphic conditions.

Contrariwise, the calcite limestone-dominated cover sediments were completely recrystallized, along with the growth of new

micas in pellitic rocks, leading to a pervasive, bedding-parallel foliation (S1). We find thrusts (SZ1) that synchronously utilized rheological weak layers of the Triassic as detachment horizons and Tertiary shales and sandstones as roof thrusts. In case of the Triassic, the cellular dolomites, evaporites and shales (Fig 4) represent mechanically weak lithologies, where strain can be easily localized upon thrusting (Pfiffner, 1993). Along these thrusts the cover sediments were detached from their substratum and formed an imbricate stack (see Sect. 4.4, Fig. 11a). Initially, this occurred by reactivation of steep SE

dipping Jurassic normal faults as reverse thrusts, which was a common mechanism for the inversion of the Helvetic shelf during the formation of the Alpine accretion wedge (Krayenbuhl & Steck, 2009; Hänni & Pfiffner, 2001). These thrusts incorporated decameter -sized slivers of basement rocks at the base of sediment stacks by the means of footwall shortcut thrusts (McClay, 1992). The shortening accumulated along each of the imbricate thrusts amounted to several kilometers.

This process most likely occurred during a final stage of the detachment of the Helvetic nappes farther to the SE and thus

during the late Kiental phase of deformation (Burkhard, 1988; Herwegh & Pfiffner, 2005; Pfiffner, 2014). This scenario clearly contradicts the interpretation of Krayenbuhl & Steck (2007) who interpreted these structures as basement folds, since (i) we do not find signs for ductile folding in the basement rock, but rather the incorporation of large slices of basement



rocks in the cover sediments; and (ii) these sediments are usually found in (thinned-out though) stacks within a normal stratigraphic succession. At the end of this deformation stage the cover sedimentary units were imbricated and stacked upon each other.

### 5.3.2 Exhumation structures

The early deformation phase did not leave a pervasive imprint in the basement rocks. However, a local basement-associated deformation is manifested by the incorporation of the early basement slivers in the sediment imbricate stack (see black arrows in Fig. 11a). This situation changed drastically during the next deformation stage. Here, the steep to sub-vertical SW-NE striking shear zones in the basement rocks developed (SZ2). They generally exhibit a reverse fault character with upward movement of the southern block (Figs. 3, 11b). Such structures can be seen through the whole Aar Massif (Baumberger et al., in press), and express the vertical extrusion of mid-crustal rocks during the Handegg phase (Herwegh et al., 2017; Wehrens et al., 2016; Wehrens et al., 2017). The large number and the dispersive distribution of these shear zones in combination with comparatively small offsets of a few centimeters to meters is characteristic and played an important role for the deformation in the cover sediments. There we find local small-scale folds (Fig. 7) with a sub vertical axial plane foliation (S2) in the JSW and just above the basement cover contact in the SE. The effect of the "Handegg" phase of vertical tectonics ended in the sedimentary cover units, where the localized shear-zones in the basement were accommodated by (deca)meter-scale folding in the mechanically weaker sediments at their contacts with the basement rock. Therefore, this deformation phase did not affect the sedimentary nappe stack at higher tectonic levels by localized shear deformation (Fig.11b). At the scale of the entire massif, however, the large-scale bulging of both, the basement cover contact and the Helvetic nappe stack is in parts related to this deformation stage.

Subsequent horizontal thrusting overprinted all aforementioned structures, producing a set of thrusts that are found both in the basement and cover rocks (SZ3). These thrusts cut into the sediments, most notable in the JSW. They further induced the formation of the mylonitic S3 foliation in the sediments, still under ductile deformation conditions for calcite. This deformation corresponds to the "Pfaffenchopf" phase of Berger et al. (2017as) and Herwegh et al. (2017). It accommodated a significant amount of horizontal displacement, and the deformation was concentrated at several levels. During this phase numerous and large slabs of basement gneisses were wedged into the sediments (see arrows in Fig. 11c). The presence of these has been known for almost a century (i.e. Scabell, 1926; Collet & Parejas, 1931; Kammer, 1989), yet their origin and particular position has never been considered as a key to understand the deformation style (Fig. 11c). These slabs document a significant amount of shortening, which accommodated during the late "Pfaffenchopf" phase of thrusting.

### 5.3.3 Youngest structures

The brittle deformation structures presented in Sect. 4.2 cut all older ones and affect the crystalline basement and the sedimentary cover alike, thus being clearly the youngest ones to be active. The steep shear zones (SZ4; Fig. 9b) do not



accommodate much offset and rather are present often as open, or partly filled, joint sets. They are strikingly similar to
structures reported by Labhart (1966). Berger et al. (2017a) described these structures in the same geodynamic context but
referred them as "Gadmen" phase structures. The (sub)vertical SE-NW running planar features reported in Sect. 4.2.1 show a
complex history of deformation, with clear evidence for brittle deformation (cataclasites of serval generations and young
fault gauges in the cores). Offsets at cm to meter scales allow to identify strike-slip to oblique fault behaviour. They have
striking similarities with faults reported from the SW Aar Massif. According to Ustaszewski et al. (2007) these offsets record
evidence for "episodic" cycles of brittle deformation and fluid pulses that formed the veins and cataclasites over millions of
years. In addition, these faults were considered to offset Quaternary sequences (Ustaszewski et al., 2007) as well. However,
both fault sets affect the crystalline basement and the sedimentary cover alike and do not feature large offsets. They are thus
not considered as great importance for the structural style and the inferred deformation history.

### 5.4 Geodynamic implications

The peak temperature estimations for the internal JSW (<330°C) indicate a depth of 11-13 km during these conditions. The
onset of dynamic recrystallization in quartz in these rocks indicates that the first deformation occurred close to these
conditions. This deformation produced the fabrics (SZ1, S1), which we link with the imbrication and the stacking of the
sediments, which was also associated with the wedging of some basement rock lenses. This deformation marked the change
from thin-skinned tectonics to thick-skinned deformation (late "Kiental" phase of Burkhard 1988) at the external European
continental margin. It is characterized by shortening and horizontal tectonics.

The drastic change in tectonic style to vertical differential uplift through reverse/normal faulting produced mainly shear
zones in the basement (SZ2) with only local folding in the sediments (with axial plane foliation S2). These structures have
been related to the Handegg phase of deformation and can be found in the entire Aar massif (i.e. Wehrens et al., 2016;
Wehrens et al., 2017; Herwegh et al., 2017) It is noteworthy that at the NW rim of the Aar massif (our study area) little offset
was accumulated. The sudden change from horizontal to vertical tectonics was induced by buoyancy forces (Herwegh et al.
2017) and was related to the rollback of the European lithospheric mantle slab and slab steepening (Schlunegger & Kissling,
2015; Kissling and Schlunegger, 2018).

Another change in tectonic style (back again to horizontal tectonics) produced the third deformation fabric (SZ3, S3). This
NW directed thrusting occurred during the "Pfaffenchopf" phase and is an expression of the remaining compressional
orogenic forces (Wehrens et al., 2017, Herwegh et al., 2017). It is during this phase that a second set of large basement rock
slabs is thrusted into the sedimentary cover. One major thrust horizon is located within our JSW, offsetting the EZ for at
least 2 km.





## 6 Conclusions

We find that linking the deformation structures in the Aar Massif basement and its cover at the Aar Massif NW rim is
aggravated by (i) the different rheological response to strain under the presented T conditions and (ii) the superposition of
several deformation structures. In this context, we first find that the key for a better understanding of the tectonic
complexities lies in the finding that $T_{max} < 330°C$ allows only for brittle deformation of feldspar, dolomite or iron-
carbonates, very limited semi-ductile deformation of quartz, and entirely ductile deformation of calcite and phyllosilicates is
key. This leads to ductile folding and thrusting in the calcite-dominated cover sediments (i.e. Upper Jurassic, Cretaceous)
while in the quartz- and feldspar-dominated basement (i.e. ILZ, EZ) semi-ductile and discrete shear zones were formed
whereas the bulk of the rock behaved in a brittle manner. Secondly, we can disentangle the imprints of at least 3 deformation
stages, each leaving different structures in the crystalline basement and the sedimentary cover. This enables us to refine the
original 2 phase-subdivision (Kiental- and Grindelwald-phase) and allows us to link our observations with the recently
published large scale block extrusion model of the entire Aar Massif, governed by the change in plate driving forces in the
lithospheric mantle. Lastly, we conclude that the multiphase tectonics oriented the basement-cover contact of the Aar Massif
in a steep NW plunging manner. The structural imprint (especially the horizontal and vertical shear zones and foliations) sets
up the stage for erosion to produce the impressive morphology of the Eiger-Jungfrau mountains.

## Data availability

Remote sensing derived lineaments used for Fig. 6a are provided as .xyz files with Swiss coordinates (CH1903) are included
in the supplement (S1).
All measured Raman spectra with intensity (in arbitrary units, second column) over Raman shift (in cm-1, first column) and
Used spectra for curve fitting (and Fig. S3 histograms) are indicated in the excel sheet, all in the supplement (S2).

## Author Contribution

DM and FS designed the study whereas DM and AL carried out fieldwork and LN did the RSCM measurements. DM
interpreted the data with additional scientific input from MH. DM prepared the manuscript and figures with contributions
from all co-authors.

## Acknowledgments

We thank the "Jungfraubahnen" Railway Company, especially Stefan Michel, for their logistic support and access to the
railway tunnel. We further thank the High Altitude Research Stations Jungfraujoch and Gornergrat (HFSJG) for making our





fieldwork possible. The research was supported by the Swiss National Science Foundation through grant No 159299

awarded to Fritz Schlunegger. Alfons Berger is thanked for discussion about basement rock units.



## Figures and Tables

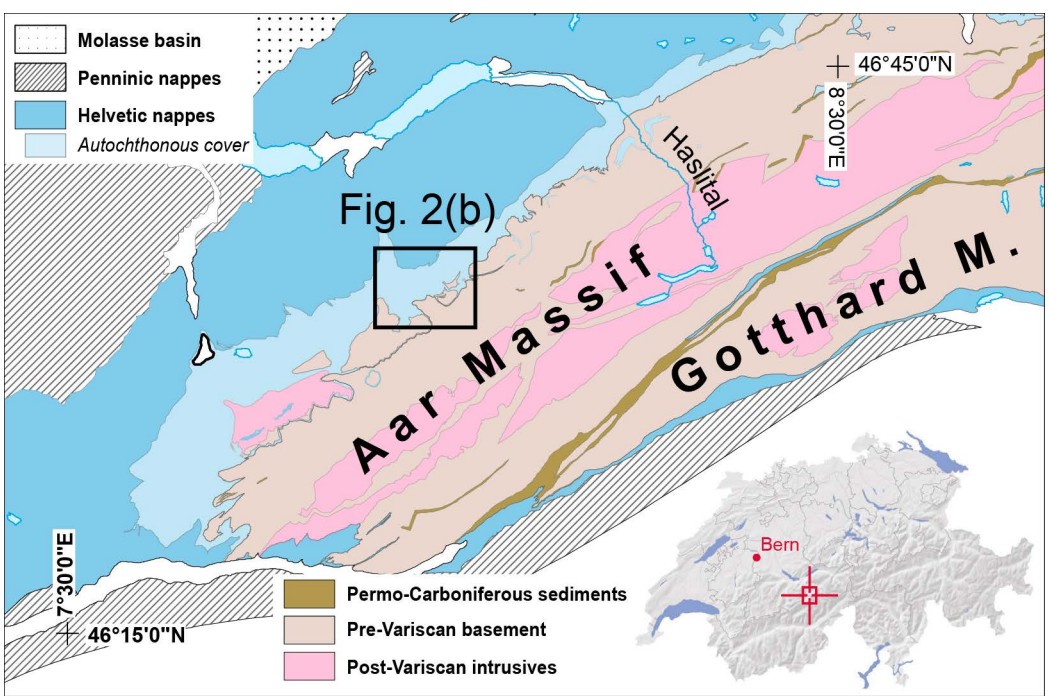

**Figure 1. Regional tectonic overview map (modified from Pfiffner et al. 2011) with location of the study area within the Swiss Alps (insert).**





**Figure 2. Structural main shear zone map based on own field work and compiled from sources as discussed in Appendix A. (a) Tectonic overview of the studied zone. (b) Refined structural and lithological map. Profile trace for Fig. 11 and key locations (EN… Eiger north, ET… Eismeer/Tunnel, EW…Eiger west, MJ… Moenchjoch, RT… Rottal, TB… Trugberg; JSW… Jungfrau sediment wedge) are indicated. RSCM sample locations are indicated (subsurface samples from the railway tunnel are indicated with dashed stars). Coordinates are given in Swiss Coordinates (CH1903).**





Figure 3. Key basement rock fabrics. Row 1: Outcrop images with indication of discrete shear zones (white arrows). Row 2: Crossed polarized light micrographs of pre-Alpine fabrics with relictic granoblastic microstructures of large feldspar and biotite crystals, and interstitial quartz. Metamorphic overprint manifests in: i) white mica and quartz growth with smaller grain sizes, ii) the alteration rims of feldspar and iii) biotite to chlorite alteration. Row 3: Shear zone micrographs. SC – fabrics are formed by white mica, chlorite and polymineralic fine-grained ultracataclasite or ultramylonite in between porphyroclasts, (which exhibit brittle deformation). Sample names are indicated; for sample details see Appendix table A1.

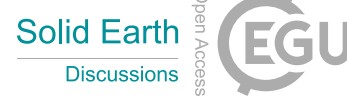

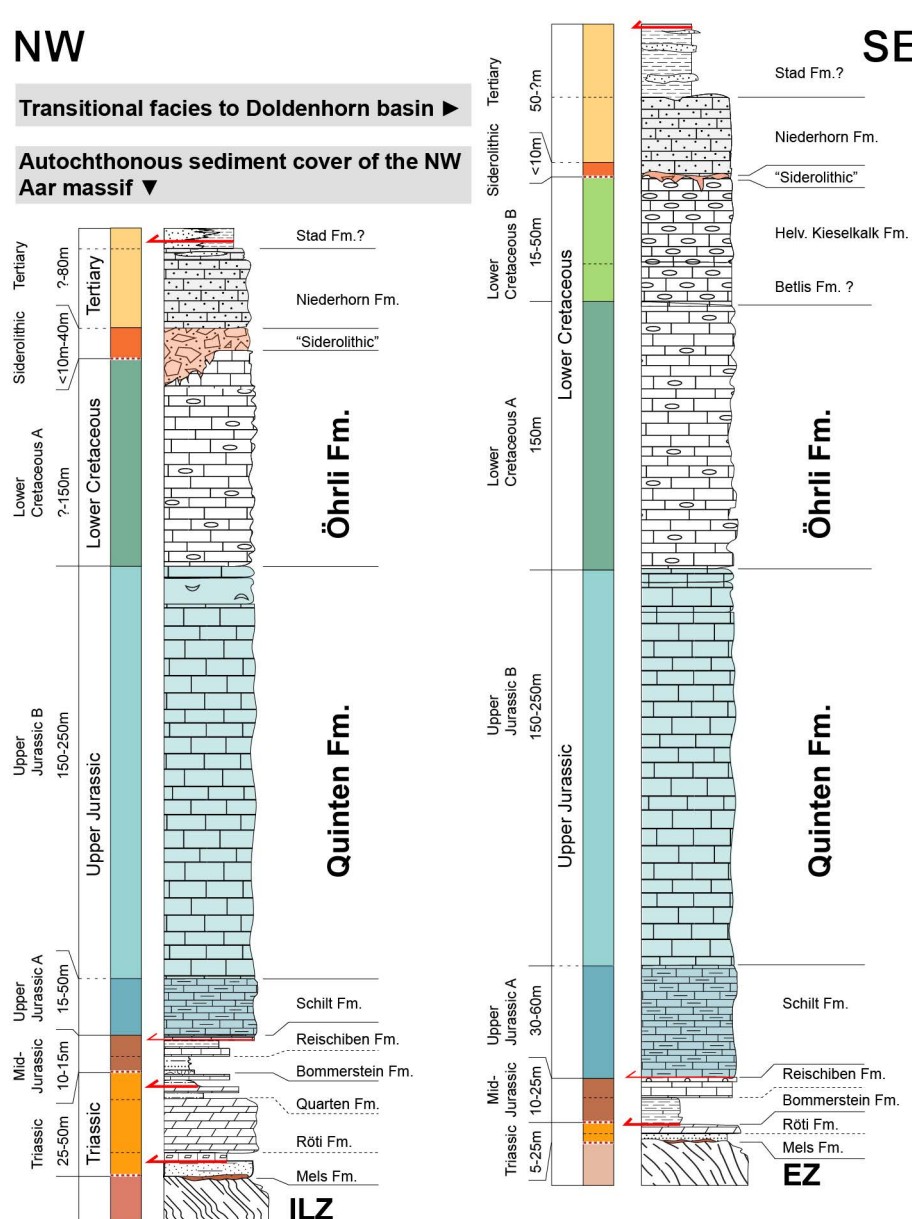


**Figure 4. Detailed stratigraphic profiles for the Mesozoic cover sediments (for sedimentological discussion see Table 2 and Appendix A). Future main detachment horizons are marked (red arrows), ILZ … Innertkirchen-Lauterbrunnen Zone, EZ … Erstfelder Zone. For references for the thickness estimates and individual unit names see table 2. This figure serves as legend for Fig. 10.**




**Figure 5. Deformation structures in the basement on the large scale in different key outcrops along the strike of the mountain chain (localities are indicated in Fig. 2). (a) Trugberg mountain ridge viewed from the west: The complex shear zone network with indication of the individual deformation phase structures Pf: Pfaffenchopf phase, Ha: Handegg phase (see also Fig. 6). (b) The Jungfrau sediment wedge (JSW) from the east with an incorporated large basement wedge that separates the Innertkirchen-Lauterbrunnen zone (ILZ) from the Erstfeldzone (EZ). (c) Late stage shear zoning below the Jungfrau demonstrating the complex cross-cutting and overprinting relationship of the different structures (Ki: Kiental phase). For deformation phase discussion and attribution see discussion in text. Location of key samples is indicated (SX-01, SX-02).**




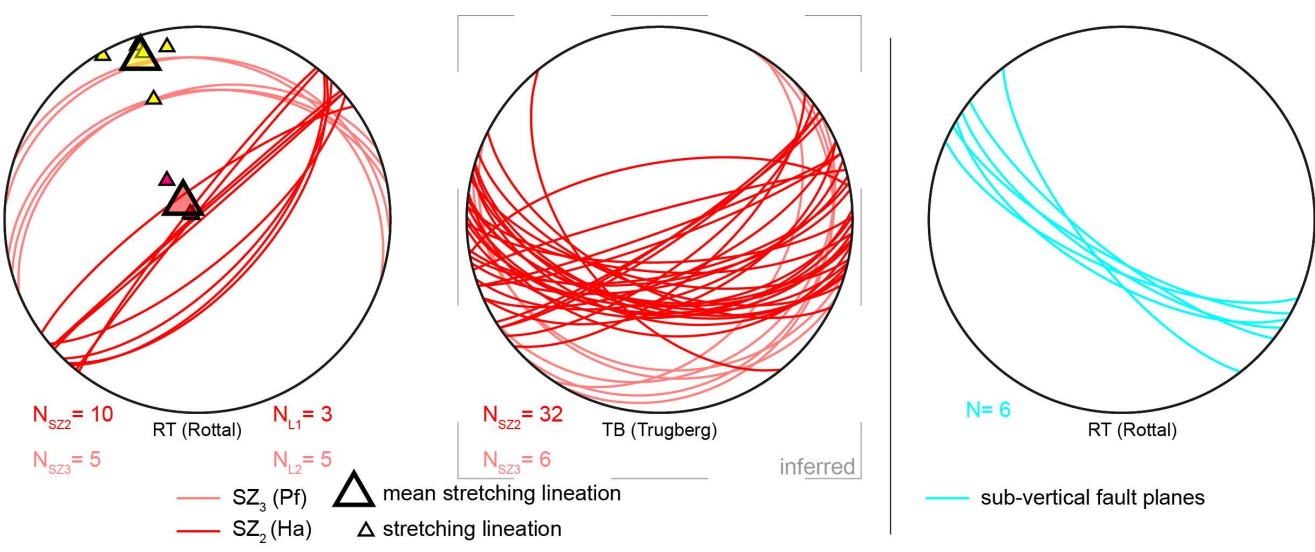

**Figure 6. Structural field data for the basement from the Rottal (RT). (a) Shear zones related to main phases of exhumation. Shear zone orientations for TB were inferred from by plane fitting through moment of inertia of remote sensed lineaments (data provided in the supplement S1). (b) Vertical lineaments that crosscut structures from (a). For geographical abbreviations and shear zone legend see Fig. 2.**






**Figure 7.** Field examples of key deformation structures within the sediments. (a) Modulation of the previous foliation (S1, white lines) by the latest stage fault system (solid red lines) and the contemporarily induced foliation (S3, pink lines) in the summit region of the Jungfrau. (b) Similar overprinting of already folded bedding by the younger foliation east of the Eiger summit in a lower level (elevation around 2500m a.s.l). (c) Decameter size dolomite boudin of the Triassic (Röti Fm.) marking the shear zone at its base in the RT (Rottal) region, which is the roof thrust of the autochthonous cover unit. (d) Decameter sized sigma clasts of boudinaged iron-rich nodules (Siderolithic Fm.) within the ductile host rock mylonites that marks a shear zone in the west flank of the Eiger Mountain. (e) Local modulation of the initial foliation and folds by steep SE plunging axial planes with foliation, inflicted during the intermediate step of deformation (Ha). (f) Carbonaceous ultramylonite in the MJ transect, some decameter above the location of (e), showing the strongly localized latest overprint, completely removing traces of previous deformation.





**Figure 8. Deformation structures on outcrop and microscale. (a) Ultra-mylonitic Tertiary sandstones intercalated with calcite limestones that (b) show a dynamically recrystallized fabric in the limestone part, while the quartz within the S-C mylonite exhibits still (semi) brittle behavior. This manifests in mylonitic ductile shear bands formed by micas and calcite while quartz grains (along with Pyrite) form sigmoidal clasts. (c) Dark and micritic limestone mylonites of the Upper Jurassic B unit with a (d) completely recrystallized fabric of microcrystalline calcite. (e) Echinoderm-rich limestone of the Mid Jurassic show the low P, T overprint with (f) un-deformed echinoderms, probably due to consisting of Mg- calcite and thus being stronger (Xu et al. 2009).**






**Figure 9. Deformation structures in the sediments. (a) Bedding parallel S1 foliation and later induced S2 foliation that consists of numerous slip surfaces. Stretching lineation are only documented on S2 surfaces. (b) Brittle-only structures that crosscut structures in (a).**




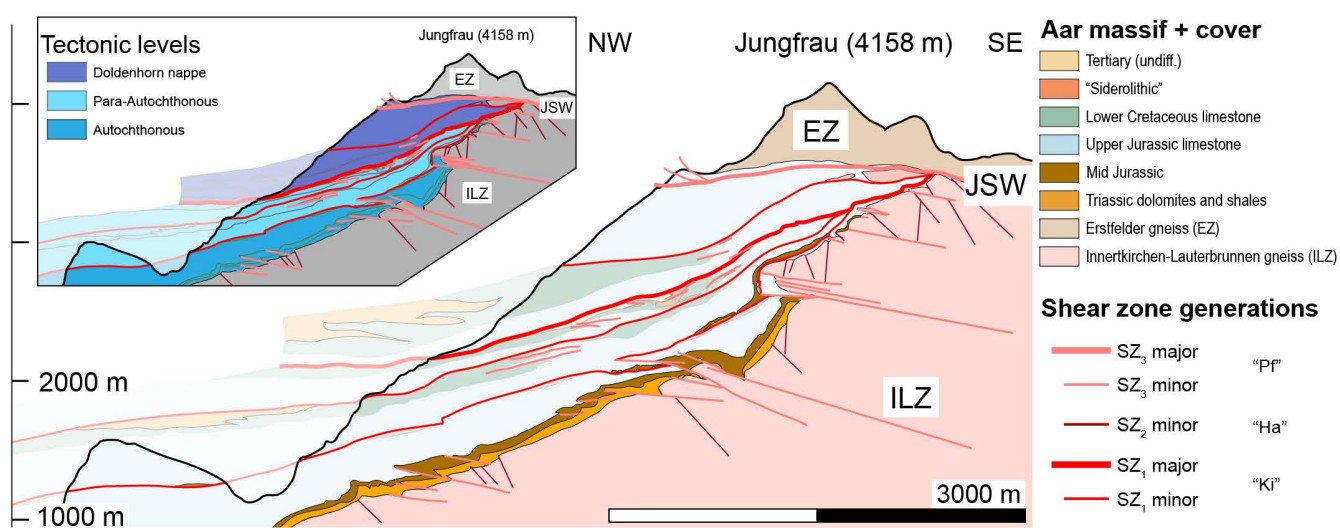

**Figure 10. Simplified structural profile across the NW rim of the Aar Massif (profile trace indicated in Fig. 2).**

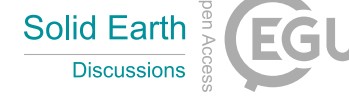

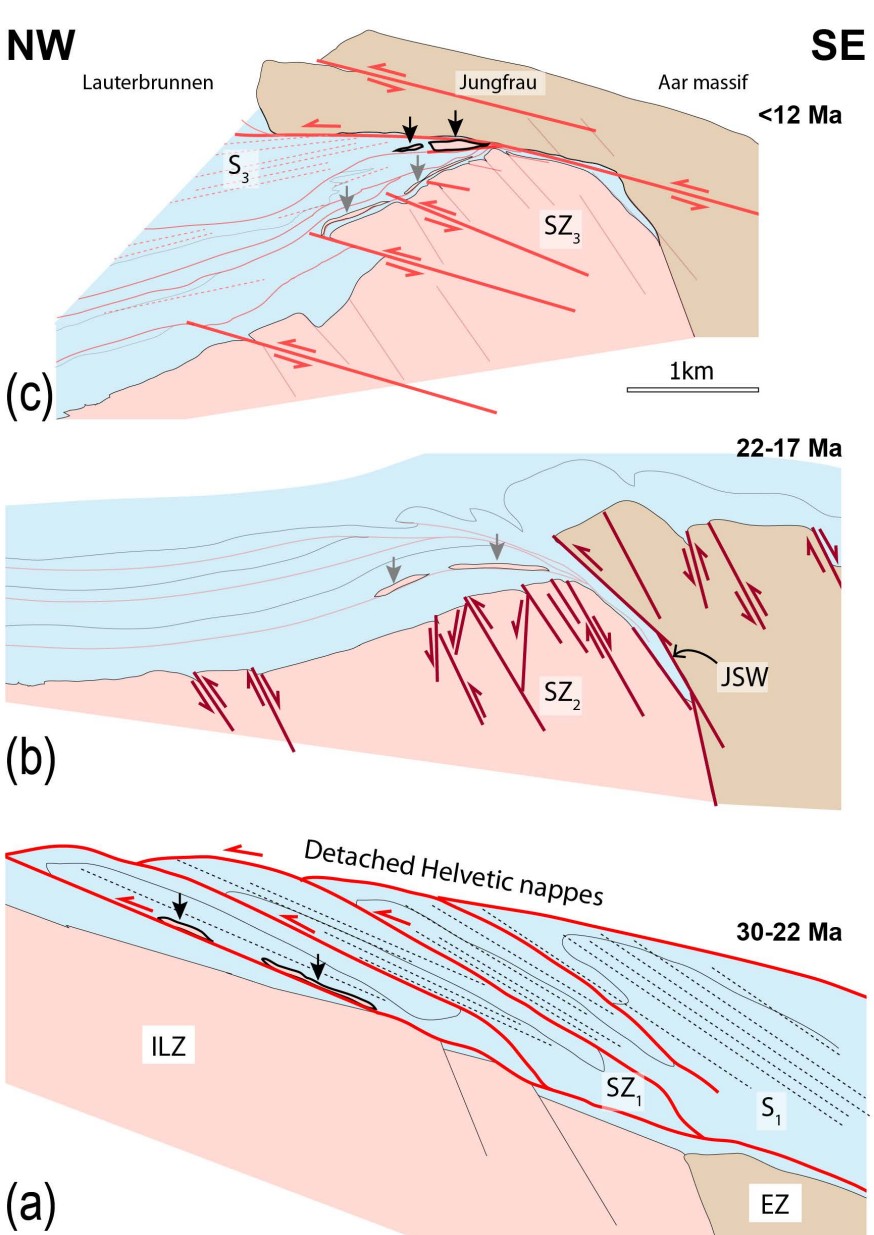

**Figure 11. Schematic sketch of the evolution during the main deformation stages of the Alpine evolution. (a) In sequence imbricate stacking during the late Kiental phase that induced S1 foliation and led to the incorporation of basement slabs into the sediment stack (black arrows). (b) Steep reverse faulting of SZ2 that induced folding in the cover during the Handegg phase. (c) SZ3 thrusts and S3 foliation in the sediments during the Pfaffenchopf phase along with second generation of basement slabs incorporated into the sediments (black arrows).**




| Age [ma] | Main phase | Sub-stage | Domain | Main characteristics | References |
|---|---|---|---|---|---|
| **30-22** | **Kiental** | | West Helvetics | Nappe stacking with folding | Burkhard (1988) |
| **20-?** | **Grindelwald** | | Central Helvetics | Passive rotation and folding | Burkhard (1988), Pfiffner (2014) |
| 22-20 | | Handegg | Aar Massif | Steep normal/reverse faults | Wehrens et al. (2016) |
| 20-17 | | Oberaar | South Aar Massif | Dextral/oblique strike slip faults | Wehrens et al. (2016) |
| <12 | | Pfaffenchopf | NW Aar Massif | Flat reverse/normal faults | Wehrens et al. (2017) |
| < 5? | | Gadmen | NW Aar Massif | Steep brittle reverse faults | Berger et al. (2017b) Herwegh et al. (in review) |

**Table 1. Compilation of deformation phases from literature.**






| Simplified layer | Fm. Name | Age | Lithology | References | Thickness constraint | Confidence | Local only? |
|---|---|---|---|---|---|---|---|
| **Tertiary** | Stad Fm. | Priabonian | Shales intercalated with sandy layers/lenses | Herb (1983), Menkveld-Gfeller et al. (2016) | | Low | no |
| | Niederhorn Fm. | Bartonian – Priabonian | Shallow marine limestones, intercalated with sandstones | Herb (1983), Menkveld-Gfeller et al. (2016) | | Medium | no |
| **"Siderolithic"** | "Siderolithic" | Lutetian – Bartonian | Erosional infill in karst pockets (sandstones, iron rich carbonates) & calcareous breccia | Menkveld-Gfeller et al. (2016) | <10m to 40m | High | no |
| **Lower Cretaceous B** | Betlis Fm./ Helv. Kieselkalk (?) | Valanginian | Brown weathering biogene spary limestone with chert layers and sandy layers in the top | Strasser (1982), this study | 50 to 90 m | Medium | yes |
| **Lower Cretaceous A** | Öhrli Fm. | Berriasian | Light grey, oolithic – biogene limestones | | ? to 150 m | Medium | no |
| **Upper Jurassic B** | Quinten Fm. | Oxfordian - Berriasian | Dark, micritic limestones; on top reef platform limestones | Collet & Parejas (1931), Masson et al. (1980) | 150 to 250 m | High | no |
| **Upper Jurassic A** | Schilt Fm. | Callovian – Oxfordian | Intercalated limestones with thin marly layers | | 10 to 50 m | High | no |
| **Mid Jurassic** | Reischiben Fm. | Aalenian – Bathonien | Echinoderm bearing calcareous breccia and Iron bearing sandstones | Bruderer (1924) | <1 to 10 m | Medium | yes |
| | Bommerstein Fm. | Toarcian – Aalenian | Shales with intercalated iron rich sandstones and echinoderm bearing calcareous breccia | Bruderer (1924) | <1 to 30 m | Medium | no |
| **Triassic** | Quarten Fm. | Late Triassic | | Bruderer (1924), this study | | | |
| | Röti Fm. | Anisian | Dolomoites: pseudomorphs after gypsum, oolithic grainstones and mudstones; well-bedded | Gisler et al. (2007), Collet & Parejas (1931), Rohr (1926) | 5 to 25 m | High | no |
| | Mels Fm. | Olenekian – Anisian | Intercalated Sandstones, clays and dolomites (partly anhydrous gypsum) | Gisler et al. (2007), Rohr (1926), this study | < 10m | High | yes |
| **n/a** | n/a | Permian? | Regolith (weathered Permian basement rock) | this study | < 5m | High | yes |

**Table 2. Key stratigraphic horizons with most important features and references.**



| Sample | x | y | Elev. [m] | stratigraphic unit | tectonic unit | RSCM-T [°C] | | 2σ |
|---|---|---|---|---|---|---|---|---|
| **MJ-01** | 643469 | 156074 | 3649 | Mid Jurassic | JSW | 308 | ± | 14 |
| **MJ-06** | 643232 | 156012 | 3744 | Upper Jurassic A | JSW | 317 | ± | 11 |
| **EN-01** | 641749 | 158733 | 2388 | Lower Cretaceous A | PA | 283 | ± | 12 |
| **Lau-02** | 636387 | 157680 | 838 | Upper Jurassic B | AUT | 283 | ± | 11 |
| **EG-17-01** | 643440 | 158638 | 3970 | Upper Jurassic B | DN | 292 | ± | 10 |
| **JT-15E** | 643351 | 157295 | 3216 | Upper Jurassic B | PA | 287 | ± | 14 |
| **AM-01** | 643859 | 157971 | 3127 | Upper Jurassic B | PA | 307 | ± | 19 |
| **GH-01** | 641095 | 156976 | 2798 | Upper Jurassic B | DN | 289 | ± | 27 |

**Table 3. RSCM results for peak metamorphic temperature estimation … Jungfrau shear zone, PA ... Paraautochthonous sediments, DN… Doldenhorn nappe, AUT … Autochthonous sediments. Coordinates are given in Swiss coordinate system, (CH1903).**




**Appendix A: Geological map compilation and Mesozoic litho-stratigraphy.**

The structural map (Figs. 1 & 10) was compiled from the following preexisting maps: Collet and Paréjas (1928), Günzler-
Seifert & Wyss (1938) and the GeoCover maps: LK 1228, 1229, 1248, 1249 which in turn largely base on the former publications. The most recent maps of Pfiffner et al. (2011) and Berger et al. (2017b), which present the geological architecture at a regional scale, were considered as well. A set of profiles produced by Collet & Paréjas (1931) and Günzler-Seifert & Wyss (1938), Herb (1983) and Hänni & Pfiffner (2001) was used as a basis. However, the first sections by Collet & Paréjas (1931) and Günzler-Seifert & Wyss (1938) have not been geo-referenced, with the consequence that the course of
the section cannot be precisely allocated. Therefore, these profiles had first to be homogenized in structural style, geo-referenced and integrated in our updated geological map.

The paleo-geographical northwestern strata (Fig. 4) has been assigned to a terrestrial flood plain environment where the sediments were directly deposited on the weathered crystalline basement (Mels Fm.). This unit is then overlain by mixed siliciclastic-carbonaceous sediments and a sequence of dolomites, thus marking a tidal flat, Sabkha-type of environment
(Röti Fm.; Gisler et al. 2007). These early Triassic sediments are preserved as quartzite, slate and dolomite (locally anhydrite bearing in the Röti and Mels Fm.). In the lowest tectonic level (= the cover of the most external part of exhumed the Aar situated in the Lauterbrunnen valley) an up to 30m thick suite of dolomites overlain with a suite of shales with up to 5 m thick dolomite beds, which have been assigned to the Röti and Quarten Fm., is preserved (Bruderer 1924). These sediments (and possibly overlying units) were subject to erosion during the Upper Triassic/Lower Jurassic. This is recorded by < 1-m
thick breccias ("Basalbreccie") containing Röti dolomite components (Krebs 1925; Frey 1968). After the hiatus, a thin succession of ferrous sandstones intercalated with echinoderm-rich limestones was deposited. This unit, which have been assigned to the Bommerstein Fm. and Reischiben Fm. and are <10 m thick at the base of the Lauterbrunnen valley (Masson et al. 1980; Collet & Paréjas 1931) also contain a thin oolithic horizon that contains iron- and manganese-rich concretions. This formation is overlain by a suite of cm- to dm-thick bedded, sandy to argillaceous limestones (Schilt Fm.).
These are gradually replaced a the dark, micritic limestone upsection, referred to as the Quinten Fm. Deposition of this latter unit commenced in the lower Oxfordian and reaching an estimated thickness between 75m to 150m in the study area (Collet & Paréjas 1931). The Quinten unit itself is overlain by fossil-rich limestones (Öhrli Fm.) of varying thicknesses. These differences in preserved thicknesses are due to a Tertiary phase of erosion where stratigraphic columns were dissected to successively deeper levels from the NW to the SW. Iron-rich sandy to argillaceous infills in karst pockets combined with a
few meters' thin horizon of iron rich sandstones are documents of this erosional phase. Related fragments, most likely of pre-Priabonian to Eocene age, are referred to as the "Siderolithikum" (Herb 1983; Wieland 1979). Locally it forms up to a 40 m-thick suite of breccias with components of Quinten-limestone, Öhrli-limestone and "Helvetischer Kieselkalk" (Wieland 1979). The overlying calcareous breccia, known as "Mürren-Brekzie" with thicknesses of up to 80 m in the Eiger north face (Günzler-Seifert & Wyss 1938; Collet & Paréjas 1931), already chronicles the Priabonian transgression resulting in the
deposition of the Niederhorn Fm. This unit is considered as equivalent of the Hohgant Sandstone Member (Menkveld-



Gfeller et al. 2016). These clastic shoreface deposits are overlain by a limestone suite referred to as the "Lithothamnienkalke" (Menkveld-Gfeller 1994). Sandstone lenses and dark bituminous carbonates (most likely Gemmenalp limestone equivalents) become more frequently upsection and grade into a succession of marls alternated with siliciclastic turbidites and calciturbidites. These sediments were mapped as "Flysch" (Collet & Paréjas 1928), but we note here they have

striking similarities with sediments in the flank of the Schwarzmönch, the depositional ages of which have tentatively been assigned to the Priabonian (Günzler-Seifert & Wyss 1938). Hence it is debatable whether the attribution to the Stad Fm. or to the North Helvetic flysch group is correct.

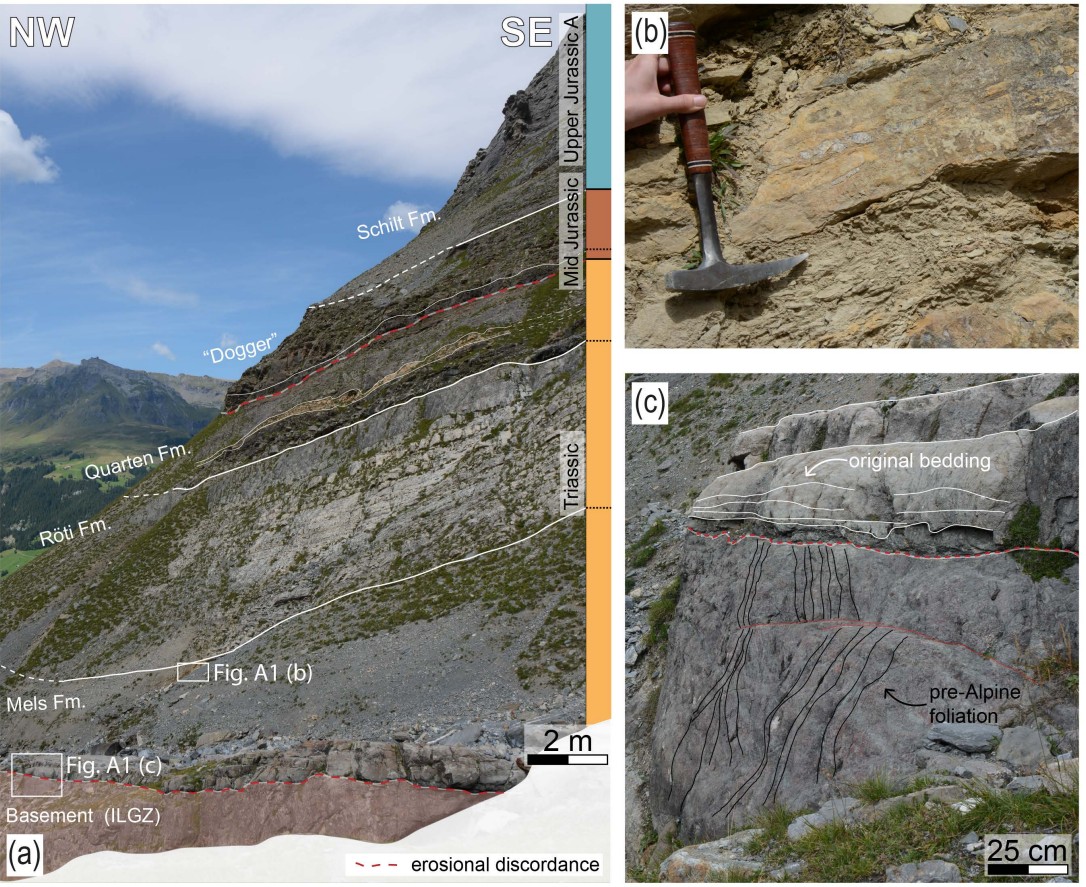

**Figure A1. Undeformed basal stratigraphic section of the cover sediments of the ILZ in the Rottal (RT).**

Within the Jungfrau sediment wedge (JSW), the sedimentary succession (Fig. 4) starts with a < 1 m-thick Permian paleosol, which has also been encountered farther to the west (Krayenbuhl & Steck 2009). This is overlain by Triassic slates and sandstones attributed to the Mels formation, and a < 2 m-thick sequence of dolomites (Röti Fm.). Similar to the situation of the Autochthon, no upper Triassic sediments are present, either due to non-sedimentation or to a phase of post-depositional erosion lasting until the Middle Jurassic (Masson et al. 1980). The succession of echinoderm-rich and iron-manganese-





nodules, deposited during middle-Jurassic times (Bommerstein Fm. and Reischiben Fm.), are considerably thicker than in the NW. This supports the inferred strong tectonic thinning of this unit (~10 m at Mönchsjoch, MJ; ~5 m at the Eismeer ET; Collet & Paréjas 1931). Similarly, the overlying sandy limestones of the Schilt Fm. are still thicker than 30 m (Fig. 3). The Upper Jurassic to Cretaceous limestones of the Quinten and Öhrli Formations are generally missing within the JSW and have been displaced to the NW. As a consequence, they can be found in the middle part of the Eiger (Fig. 2), above the

Eismeer/Tunnel (ET) area and in the northern flank of the Mönch (Fig. 2). The entire sedimentary stack is strongly folded, foliated and thrusted, which leads to a doubling and to an inversion of the upper part, while the succession has been repeatedly stacked in other localities.

| Sample | x | y | Elev. | rock type | Domain | Qtz rec. | mica dominance | Mica in SZ | Def. structures | SZ generation |
|---|---|---|---|---|---|---|---|---|---|---|
| JT-23 | 642048 | 155392 | 3388 | gneis (polymetamoprh) | ILZ (tunnel) | n/a | WM>Chl>Bt | n/a | n/a | n/a |
| JT-27 | 641960 | 155313 | 3417 | gneis (polymetamoprh) | ILZ (tunnel) | n/a | WM, Bt | n/a | n/a | n/a |
| MJ-04 | 643499 | 156091 | 3647 | gneis mylonite | ILZ | BLG | Chl | WM | mica shear bands | SZ3 |
| MJ-05 | 643232 | 156012 | 3744 | gneis mylonite | JSW | BLG | Chl | Chl | mica shear bands | SZ3 |
| SX-01 | 641944 | 155292 | 3565 | gneis mylonite | EZ | BLG | WM, Chl | WM | mica shear bands | SZ2 |
| SX-02 | 641944 | 155292 | 3565 | gneis (polymetamoprh) | EZ | BLG | WM | n/a | n/a | n/a |
| GH-01 | 641095 | 156976 | 2798 | Calcite/sandstone mylonite | Tertiary | n/a | WM | WM | Cc regrowth; SC, fabrics | S1, S3 |
| JT-16E | 643351 | 157295 | 3215 | Calcite mylonite | Upper Jurassic A | n/a | n/a | n/a | Cc regrowth | S1, S3 |
| MJ-01 | 643469 | 157295 | 3215 | Echinoderm breccia | Mid Jurassic | n/a | n/a | n/a | SC fabrics | S1, S2, S3 |

**Table A1. Basic data for selected samples used in Figs. 3 & 8. Thin section description for dominant dynamic quartz**
**recrystallization mechanism (BLG… bulging), dominant mica and mica growth in shear zone (if applicable). For shear zone generation discussion see Sect. 4.2.1.**



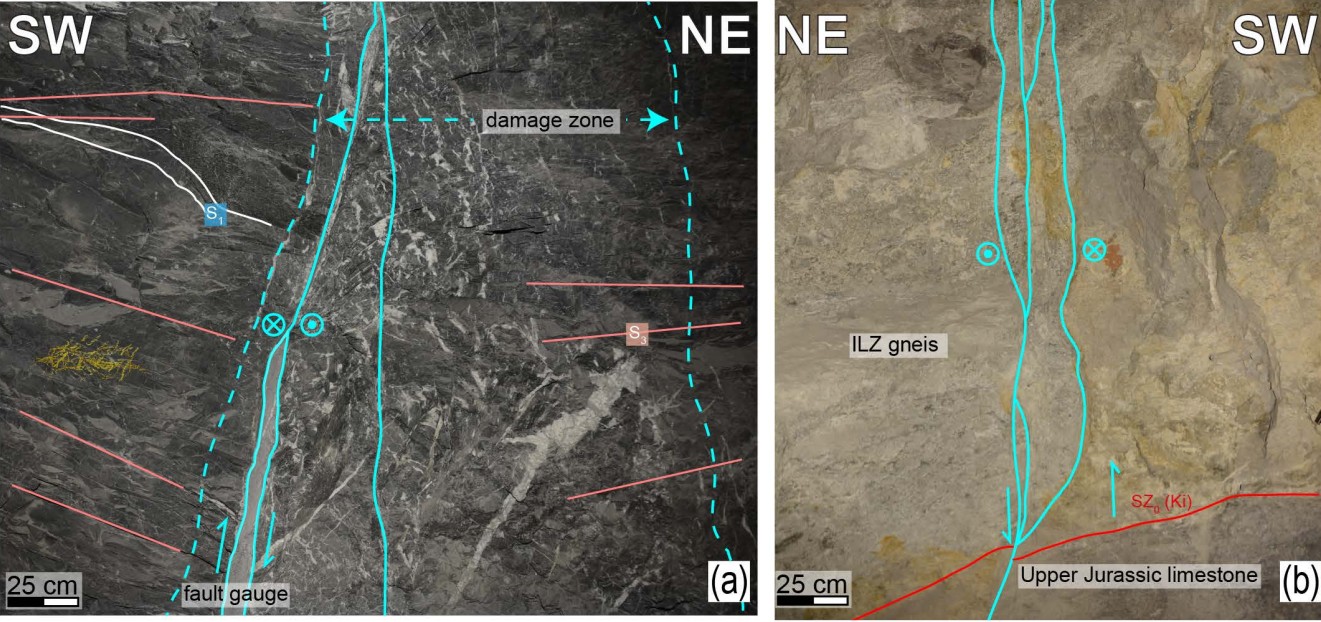

**Figure A2. Youngest structures cross-cutting all previous structures in the sediments (a) and the basement (b) with an oblique to strike-slip behavior.**



## Appendix B: RSCM temperature estimate histograms

Fitted Raman spectra distributions (Fig. B1) display reasonable gaussian probability distributions with max. spread of 50°C (except for GH-01).

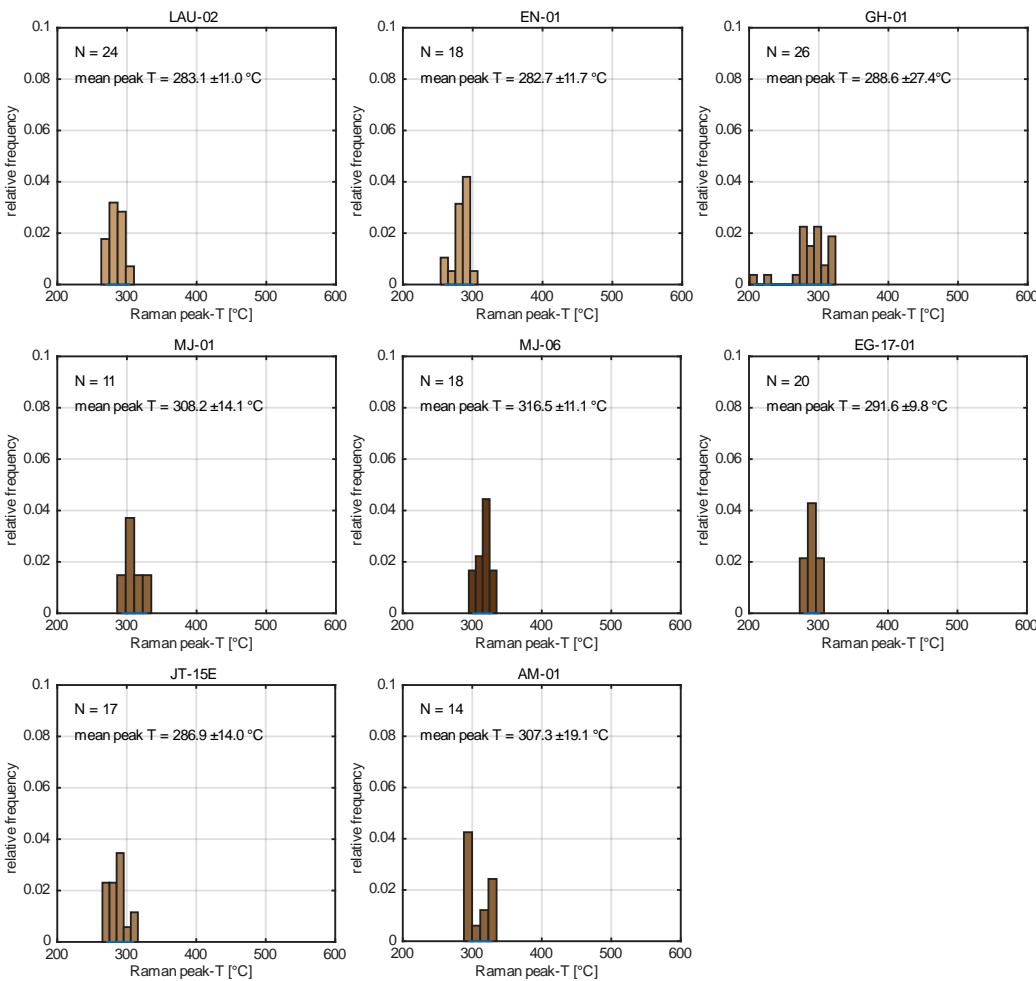

**Figure B1. RSCM temperature estimate histograms for each sample for the fitted spectra.**



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
