# Peer review of "Linking Alpine deformation in the Aar Massif basement and its cover units – the case of the Jungfrau-Eiger Mountains (Central Alps, Switzerland)"

_Solid Earth, 2018_

## Referee Comment (RC1) · Anonymous Referee #1 · 2 Jul 2018

This paper presents a detailed structural study of the northwestern Aar Massif. The text is beautifully illustrated by 11 excellent figures in addition to an appendix, but I have difficulties with the detailed distinction of numerous deformation and fabric events and I am not sure that they are of interest for a large, international audience. I believe that the valuable data and interpretations of this paper should be presented to a more regionally-focused, Alpine journal. This is also reflected by the conclusions of the paper whose essence is that the observed structures are consistent with a previously published model. The distinction between phases of deformation in the basement, based

on their orientation doesn't seem to be very robust, especially between SZ2 and SZ3 (see detailed comments). Their different orientations could reflect a large scatter of the structures rather than two distinct populations.

SPECIFIC COMMENTS Lines 12- 13 : replace Âń T at deformation ranged from xx. . . Âż with : "Deformation temperatures range between xx and xx. . ." Line 14: either "ductilely" or "in a ductile manner". Here you are using the present tense, on line 13 it's the past. . . Line 21: "NW-striking" is enough Line 22: "feature an immense topographic expression": a complex, but not very clear sentence to say "high altitude"? Line 22: replace "SE-NW striking rim" with "NW striking rim" Line 23: "offset" between what and what? Line 24: replace "the sedimentary" by "its sedimentary" and delete "rocks of the Aar Massif". Line 28: Helvetics Line 29: early stage of what? Line 29: you can decouple from a basement but you cannot decouple from an evolution (at least not in this context). Line 32: add a 150 years old reference Line 36: later? Line 39: delete "which" Lines 40 to 43: there is no real usable information for the readers here. Line 49: delete "and to fill the knowledge gap". Line 53: delete "in the field and" Line 56: Delete "In addition. . ..history". Line 62: Repetition of line 20 Line 66: "in between": not very clear Line 68: SW-strike Line 75: delete "Alpine" and add "of Alpine age" after "metamorphism". Line 84: Replace "These mechanisms are considered not to. . ." with " This process does not appear to be" Line 85: Massif Line 86: add reference at the end of sentence Line 86: what sort of deformation fabric? Line 88: the name of the deformation phase is not so interesting if the kinematics of the deformation are not described. Line 90: "thrusting": nw-vergent? Line 93: which recumbent fold? Does it have a name? Is it visible in a figure? Any references? Line 95: what is an inverse succession? Line 100: I would delete "differential" Line 101: delete "a phase of" Line 101: "strain partitioning": only interesting to know if you describe what is partitioned into what Line 101: "simultaneously with dextral strike-slip": unclear. Is dextral strike slip part of the partitioning? Lines 103-104: Pfaffenkopf and Oberaar have no references Lines 104-105: delete "the uplift passively rotated". The uplift is not a force. . . And add "were passively rotated" after " Doldenhorn Nappes". Line 108: which type of brittle

structures? Line 114: not necessary Line 117-118: delete Line 120: "only" should be before, not after mapped Line 121: delete only Line 145: basement Line 145: replace "are present as" with "consist of" Line 148: what is a "granodioritic texture"? Magmatic texture? Line 152: delete "see also" Line 155: evidence Lines 174-176: delete sentence Lines 180: brittle shear zone? Lines 181-182: "Note that....sediments" doesn't need to be between parentheses. Line 182: replace "faulting behavoiur" with kinematics Line 183: "However" ??? Line 183: I cannot see the offset of older structures in Fig. 5b Line 189: no dot after SZ2 Line 189: "offsets": this could give you a shear sense? Why is there no shear sense described? Looking at Fig. 6 I do not find the distinction of SZ2 and SZ3 convincing based on the orientation data. The Rottal net shows a possible distinction into two groups based on the different dip, but there aren't many data and even there it could be one population only. The Trugberg net doesn't show two"populations" in my view. Line 190: "wider": give an idea Line 190: replace "occurred" with "is" Lines 190-193: any references to figures? Line 193: "kinematics": you did not say much about it in the last lines (see comment on Line 189) Lines 201-204: a verb is missing in this sentence Line 210: I doubt it....Mapping certainly reveals something, but I guess something else. Line 213: why "the first"? It was not mentioned that there were many. Is it S1? Line 214: what is the spacing of mylonitic foliations? Spacing between shear zones? Line 217: The reference to Fig. 7c comes after "more than one def phase", but it doesn't show that. It only shows the boudinage. Line 219: past tense (led) not always used in the text. Reference to maps and figures? Line 222: argument for the synchronous development of S1 and SZ1? Line 223: what is the shear sense of these shear zones? Line 225: it needs to be stated before that these shear zones are thrusts. And what displacement direction? Line 225: "of" the footwall rather than "in"? Line 227: subsequent to what? Line 228: what is a rotated sigma clast? And...why does a rotated sigma clast indicate rotation of the initial structures? And ... what deformation phase causes an anticlockwise rotation of some 50°, looking East? "As a result": of something that is not really explained. Line 229: what are EN and EW? Line 230: "This deformation stage": I am lost. Which one? The one that rotated the

fabrics? What are the kinematics of this deformation? Line 230: how do you know it is not preserved? Line 231: "and the folding": so this deformation phase is the folding? What axial plane and fold axes orientation? Line 233: "Subsequent NW directed shearing": not easy to understand and visualize and I cannot really see it in Fig. 11b that is quoted at the end of the sentence. Where do I see the flat-lying limbs? What is the evidence for a subsequent NW-directed shearing in the first place? Line 234: "now": do you think the limbs were initially steep? Line 234: this observation of the result should be described by a photo or sketch of the structures, not by a schematic recap of the Discussion/Conclusions. Line 234-235: S3? How can one distinguish S3 from S2 that are sometimes parallel? Line 236: difficult to follow. Why is it really necessary to distinguish an S3 from an S2 here? Line 237: which one is the "same" orientation? Line 237: "This foliation...cut": I don't understand the sentence. Lines 210-240: not easy to follow, especially the mixture of orientation with cross-cutting relationships and of microstructural characteristics. Line 243: what does oblique mean here? Kinematics of these faults? Line 258: specify and justify the geothermal gradient used for the conversion of T in depth Line 266: evidence for being deformed as an "ensemble"? Line 267: consistent? With what? Line 268: less shortening: evidence? Quantified? Lines 269-270: why? Line 272: The cover-sediment interface? Line 283: why "already"? Line 284: has this wedging been described before in the text? Wedging associated with folding? I am lost... Line 286: evolution of what? Line 289: succeeding??? Line 293: favorable for what? Lines 295-301: I am not sure why these observations are important in the context of the results session and of the paper. Lines 302-305: why to mention an own stratigraphic model here that was not presented in the results? Lines 305-307: I am still not sure about the importance of these lines. Line 311: Substratum T constrained by RSCM? Line 311: why to give an upper limit to calcite thermometry and mention calcite thermometry at all? Line 314: ok, but you should also quote the lower T suggested by XX and White. I think it must be Burkhard, 1993, not Burkhard, 1990. Line 322-323: repetition of line 317. Line 323: well, I am not sure that this can be constrained. The only thing that can be said is that T of 270° C was attained following

the recrystallization T inferred by Stipp et al., 2001. What is known about the T peak anyway? Line 325: pelitic Line 325: synchronously: what is the evidence? Line 326: delete "rheological" Line 333: was it quantified? What is the evidence? Line 344: was really shown that incorporation of basement slivers in the cover is associated to the 1st deformation phase? Line 361: how significant? Line 365: repetition of 361 Line 369: so why should they be called shear zones? Line 380-384: So S1 is at < 330°C, shows dynamic rexx of quartz, but no fabric in the basement, as stated in Line ....? Lines 386 – 388: so ... under which T conditions did these shear zones develop? Lines 390-392: An amazing change of scale of interpretation! Lines 394-395: 3remaining compressional orogenic forces": I think these speculations are not really necessary. Line 400: "aggravated": aggravating a link? First, it sounds quite dramatic, second it's not the link, but its interpretation that may be "aggravated". Line 404: "is key": structure of the sentence needs to be reconstructed. Line 405: "and discrete":? Lines 404-406: "while" and "whereas" is too much for the same sentence. Needs to be reformulated. Line 406: "bulk of the rock behaved in a brittle manner": do you mean that between the shear zone the rock was also deforming but in a brittle manner? Not clear. Line 410: "multiphase tectonics": strange term. Lines 411- 412: "the structural imprint ....sets up the stage for erosion": strange statement. I guess the authors wish to say that the steeply oriented displacements created uplift and exhumation by erosion?
* * *

---

## Referee Comment (RC2) · F. Neubauer (Referee) · 10 Jul 2018

General remarks The manuscript is dealing with structures formed at peak conditions of very low-grade to low-grade metamorphism at the NW margin of the Aar Massif. Although the area is highly challenging for geological work, these structures are well documented and a succession of structures is established, which are related in part to established phases. The arguments for timing of all these phases are not well constrained in the manuscript itself, particularly for readers outside of the Alpine community. Consequently, I recommend add a few sentences on which geochronological and/or sedimentary data the timing of these phases is based. The manuscript is well written, the data are after my knowledge new and contribute to the interpretation of root zone of Helvetic nappes. I recommend publication after some minor to moderate revision, the new data warrants publication.

Specific remarks (see also enclosed annotated pdf-file with further remarks and few corrections of typos). General: use "sedimentary rocks" instead "sediments" (which is rather Central European use). Line 15: …. multiple deformation stages before and during the Aar Massifs rise: You mean rise or exhumation? Line 18: vertical or horizontal block extrusion models? Line 21: Looking foer Fig. 1b, it is difficult to recognize the NW-SE strike of teh frontal margin. The overall margin seems rather NE-SW trending. Line 24: be specific: pre-Alpine (Variscan) crystalline substratum Line 52: synthesized lithostratigraphic framework Line 124: Show trace of the "Jungfraubahnen" railway tunnel in Fig. 2. I could not find it. Line 234: Why the jump suddenly to the model Fig. 11b? Line 329: ould you show these reactivated normal faults resp. thrusts on Fig. 2? Line 353: weaker sedimentary rocks than what other rocks? Figure 1: Why post-Variscan intrusives. It seems rather late-stage Variscan. Fig 3: Explain mineral abbreviations in Rows 2 and 3. i) These two generations of minerals are difficult to recognize. Fig. 8: Explain mineral abbreviations. Use crinoid or echinodermata instead of echinoderm. Fig. 11: Also indicate the potential root of the incorporated basement slabs in (a). Table 2: The table needs some corrections of typos. Furthermore, explain in caption what you mean with "Confidence" and "Local only?" Table A1: The table needs some corrections of typos. Appendix A: Geological map compilation and Mesozoic litho-stratigraphy: Some corrections are needed, too.

Please also note the supplement to this comment:
https://www.solid-earth-discuss.net/se-2018-49/se-2018-49-RC2-supplement.pdf

**Supplement:**

[revised manuscript text omitted]

---

## Author Comment (AC1) · 9 Aug 2018

We thank the reviewer for the constructive and insightful comments, and the general positive review of our work ("Although the area is highly challenging for geological work, these structures are well documented and a succession of structures is established, which are related in part to established phases. [. . .] The manuscript is well written, the data are after my knowledge new and contribute to the interpretation of root zone of Helvetic nappes."). We carefully addressed all comments and followed the suggestions to improve the manuscript (see line by line response in the Supplement to this comment). We gladly follow the reviewer's recommendation to provide more discussion on the geochronological constraints of the deformation phases to appeal to a broader audience ("The arguments for timing of all these phases are not well constrained in the manuscript itself, particularly [. . .] Consequently, I recommend add a few sentences on which geochronological and/or sedimentary data the timing of these phases is based."). We do so by weaving the methods used for in the literature in Sects. 1 and 2, and by expanding the scope of the discussion Sect. 5.4, where we now discuss the chronological constraints on the deformation phases. To those we only we link our relative deformation patterns, as we did not produce chronological data, and the scope of the paper is to present the new and spectacular structures and their relative chronology.

For our line-by-line response we refer to the Supplementary of this comment and for the changes made to the manuscript we refer to the Supplementary to the response to reviewer comment 1.

Please also note the supplement to this comment:
https://www.solid-earth-discuss.net/se-2018-49/se-2018-49-AC1-supplement.pdf

[Figure]

**Supplement:**

**Authors response to Reviewer comment 2 (by Franz Neubauer)**

David Mair[1], Alessandro Lechmann[1], Marco Herwegh[1], Lukas Nibourel[1], Fritz Schlunegger[1]
[1] Institute of Geological Sciences, University of Bern, Baltzerstrasse 1+3, CH-3012 Bern

*Correspondence to*: David Mair (david.mair@geo.unibe.ch)

**Line by line response**

*General comment 1: General: use "sedimentary rocks" instead "sediments" (which is rather Central European use)*

Response: Done.

*Line 15: multiple deformation stages before and during the Aar Massifs rise: You mean rise or exhumation?*

Response: Was changed to exhumation to be consistent.

*Line 18: Line 18: vertical or horizontal block extrusion models?*

Response: It is vertical block extrusion; However, the sentence has now been completely removed to increase clarity and conciseness.

*Line 21: sedimentary rocks*

Response: Changed (see also general comment 1).

*Line 21: Looking for Fig. 1b, it is difficult to recognize the NW-SE strike of the frontal margin. The overall margin seems rather NE-SW trending.*

Response: Corrected.

*Line 24: Line 24: be specific: pre-Alpine (Variscan) crystalline substratum*

Response: Changed accordingly (without the "Variscan" to avoid confusion over the quite complex pre-Alpine evolution of the basement).

*Line 52: synthesized lithostratigraphic framework*

Response: Suggested change was made.

*Line 91: and now Lower Helvetic*

Response: Changed as suggested.

*Line 94: at around 20 Ma*

Response: Implemented.

*Line 116: basis*

Response: Corrected.

*Line 124: Show trace of the "Jungfraubahnen" railway tunnel in Fig. 2. I could not find it.*

Response: The railway tunnel is indicated on the map (semitransparent green line) and also shown in the map legend. Therefore, we refrained from further highlighting it as we feel it would be otherwise visually overrepresented in the map.

Line 161: *Fms.*

Response: Corrected.

*Line 166: rather lithologic than stratigraphic content*

Response: Changed accordingly.

*Line 179: exhumation-related*

Response: Corrected.

*Line 234: Why the jump suddenly to the model Fig. 11b?*

Response: The idea is to illustrate the sediment imbrication and we now reference Fig. 10, since it is better suited to do so, and it keeps the figure referencing consistent.

*Line 262:* superficial words deleted as suggested.

*Line 297: towards the paleo-SE,*

Response: Corrected, as suggested.

*Line 300: Somehow unclear: Deep water conditions in the NW when the normal faults trend NW-SW?*

Response: Clarified by changing it to "paleo-SE", to avoid confusion as these sediments are presently found to the SW of our study area.

*Line 306: Table*

Response: Corrected.

*Lines 318 and 319*; mineral names corrected

*Line 325:* corrected to "pelitic"

*Line 329: Could you show these reactivated normal faults resp. thrusts on Fig. 2?*

Response: These normal faults are hard to indicate, as they are strongly overprinted by the subsequent stages. The main such fault array is the JSW, which then later is a major SZ3 thrust and therefore indicated as such. Thus, unfortunately we see no better way to indicate those normal faults in another way than we already did.

*Line 353: weaker sedimentary rocks than what other rocks?*

Response: We named the relevant units and added a reference to the section where we discuss the sediments in question.

*Figure 1: Why post-Variscan intrusives. It seems rather late-stage Variscan.*

Response: The timing and geodynamic context of those units is quite complex, we wanted to follow the map of Berger et al. (2017b) but mislabeled them. We corrected it to "Late to Post-Variscan intrusives".

*Fig 3: Explain mineral abbreviations in Rows 2 and 3. i) These two generations of minerals are difficult to recognize.*

Response: We added mineral abbreviation descriptions in the figure caption. To better illustrate the 2 mineral generations, we increased the magnification of the first and the third image in the second row. We highlighted a core rim structure stemming from the later overprint and labelled the preserved high-T fabric differently (now black) compared to the later grown minerals (now red). The difficulty in illustrating is that we want to highlight the fabric without preferred orientation. By zooming in too much (for better depicting the smaller grains of the later growth) this is not comprehensible anymore. We hope the revised figure now offers an acceptable compromise between both.

*Fig. 8: Explain mineral abbreviations. Use crinoid or echinodermata instead of echinoderm.*

Response: We added mineral abbreviation descriptions in the figure caption and changed "Echinoderms" to "Echinodermata". We corrected the figure caption following the suggestions.

*Fig. 11: Also indicate the potential root of the incorporated basement slabs in (a).*

Response: We made changes to (a) to indicate the most-likely origin.

*Table 2: The table needs some corrections of typos. Furthermore, explain in caption what you mean with "Confidence" and "Local only?"*

Response: The "Confidence" was a qualitative assessment of our confidence in a) the Fm. attribution and b) the thickness constraint derived from the cited sources. The "Local only" column

was supposed to indicate the continuity throughout the study area. Much of this is discussed in Appendix A. Thus, we proceed to remove the two columns to increase comprehensibility of the table 2. We now reference Appendix A in the figure caption for additional information. All errors within the table have been corrected following the suggestions.

5    *Table 3: Paraautochthonous*

Response: Corrected to "Para-Autochthonous"

*Table A1: The table needs some corrections of typos.*

Response: All corrections were made following the reviewer's suggestions.

*Appendix A: Geological map compilation and Mesozoic litho-stratigraphy: Some corrections are*
10  *needed, too.*

Response: Corrections were made following the comments (and highlight in the marked-up pdf; see also response to comments on Lines 512 to 522 below).

*Line 512: northwestern strata of which age?*

Response: late Permian to early Triassic; text is now amended.

15  *Line 515: Lower*

Response: Corrected.

*Line 519: Late Triassic/Early Jurassic*

Response: Corrected.

*Line 522: Reischiben Fm., and*
20  Response: Corrected.

---

## Author Comment (AC2) · 9 Aug 2018

We thank the anonymous Referee for the constructive, insightful and detailed comments, which were very helpful to improve the manuscript. We did so by carefully addressing all the comments and following all suggestions (see line by line response below). In particular, Reviewer 1 raises 2 major points: First, the distinction of the deformation phases ("[. . .] difficulties with the detailed distinction of numerous deformation and fabric events [. . .]" and "The distinction between phases of deformation in the basement, based on their orientation doesn't seem to be very robust, especially between

SZ2 and SZ3 [. . .]"). We base the distinction on more criteria than just the orientation (i.e. cross-cutting relation, spatial distribution of thrusts; as already pointed out in the original manuscript). We thus see the need to clarify our observations and argumentations, where we present the deformation structures (Sect. 4.2 mainly). We address this by improving the aforementioned section and additionally consider the comments by reviewer 1 regarding Lines 183 to 266. We provide now a clearer and more detailed description of the cross-cutting relationships, orientation and field-based data that led us to the subdivision of the deformation stages. In doing so, we add a new figure (now Fig. 10) to better illustrate not only the crosscutting but also overprinting relationships (following reviewer 1's suggestion; see comments on Lines 210 to 240). Second, reviewer 1 states "that the valuable data and interpretations of this paper should be presented to a more regionally-focused, Alpine journal. This is also reflected by the conclusions of the paper whose essence is that the observed structures are consistent with a previously published model." We appreciate the general positive perception of our work. As for the relevance, we emphasize that although much work has previously been published on the different deformation models ("Grindelwald phase"; i.e. Burkhard, 1988; vertical versus horizontal exhumation, Herwegh et al., 2018), we document the first time how the exhumation processes affected the sedimentary cover and the crystalline basement in a different way. In particular, the first phase of thin-skinned tectonics affected the sedimentary cover only, while Alpine tectonic structures in the basement were related the subsequent thick-skinned deformation and the exhumation of the Aar massif only. To enhance the readability for readers working outside of the Alps we follow the recommendations of Reviewer 2 and discuss the more general framework for the Aar massif exhumation and the specific significance of the discussed evolution in the discussion (Sect. 5.4).

For our line-by-line response and the changes made to the manuscript we refer to the Supplementary of this comment.

Please also note the supplement to this comment:

https://www.solid-earth-discuss.net/se-2018-49/se-2018-49-AC2-supplement.pdf

[Figure]

**Supplement:**

**Authors response to Reviewer comment 1 (by anonymous Referee)**

David Mair[1], Alessandro Lechmann[1], Marco Herwegh[1], Lukas Nibourel[1], Fritz Schlunegger[1]

[1] Institute of Geological Sciences, University of Bern, Baltzerstrasse 1+3, CH-3012 Bern

5 *Correspondence to*: David Mair (david.mair@geo.unibe.ch)

**Line by line response**

*Lines 12- 13 : replace Á´n T at deformation ranged from xx: : :Â˙z with : "Deformation temperatures range between xx and xx…"*

Response: This has been implemented.

10 *Line 14: either "ductilely" or "in a ductile manner". Here you are using the present tense, on line 13 it's the past*

Response: Both errors were corrected as suggested.

*Line 21: "NW-striking" is enough*

Response: Indeed; changed to "NE striking" to correct also mislabelling.

15 *Line 22: "feature an immense topographic expression": a complex, but not very clear sentence to say "high altitude"?*

Response: This has been clarified. (see next 2 comments).

*Line 22: replace "SE-NW striking rim" with "NW striking rim"*

Response: done.

20 *Line 23: "offset" between what and what?*

Response: Offset between the surface elevation. Wording changed to clarify the sentence.

*Line 24: replace "the sedimentary" by "its sedimentary" and delete "rocks of the Aar Massif".*

Response: Changed as suggested.

*Line 28: Helvetics*

25 Response: Error corrected.

*Line 29: early stage of what?*

Response: Early stage of the Alpine evolution. Sentence was adjusted accordingly.

*Line 29: you can decouple from a basement but you cannot decouple from an evolution (at least not in this context).*

30 Response: The Alpine evolution of the Helvetics was decoupled from the Alpine evolution of the massifs basement rocks. The sentence and the following sentence were restructured to correctly convey this message.

*Line 32: add a 150 years old reference*

Response: There are several works from the late 19[th] century (i.e.by Escher v. der Linth, 1837;
35 Baltzer, 1880;). They are discussed extensively in the cited references; thus, they are not only cited now, also "references therein" is added to already cited works.

*Line 36: later?*

Response: Later In the sense of 2nd half of the 20th century. Sentence was restructured to avoid time confusion.

*Line 39: delete "which"*

Response: Replaced with "that" in order to correct sentence.

*Lines 40 to 43: there is no real usable information for the readers here*

Response: The sentence lists the main works over the recent years on the deformation and its chronology in the massif, which we feel should be cited in this paragraph, designed to give a brief overview of previous work. We did expand the sentence to specifically provide some information about the methods used by the studies to advance the understanding of the deformation history of the Aar massif.

*Line 49: delete "and to fill the knowledge gap".*

Response: done.

*Line 53: delete "in the field and"*

Response: Done; replaced with "on the surface" to clarify that not all samples are from the railway tunnel.

*Line 56: Delete "In addition…history"*

Response: Followed to make the sentence more concise.

*Line 62: Repetition of line 20*

Response: The reoccurring part was deleted to avoid redundancy.

*Line 66: "in between": not very clear*

Response: Clarified by adding "those gneiss units".

*Line 68: SW-strike*

Response: done.

*Line 75: delete "Alpine" and add "of Alpine age" after "metamorphism"*

Response: done.

*Line 84: Replace "These mechanisms are considered not to…" with " This process does not appear to be"*

Response: done.

*Line 85: Massif Line 86: add reference at the end of sentence Line 86*

Response: Reference was added.

*Line 86: what sort of deformation fabric?*

Response: We do clarify the phrasing here: The mentioned phase was characterized mainly by thrusting. The sentence now reflects this (see also next comment).

*Line 88: the name of the deformation phase is not so interesting if the kinematics of the deformation are not described.*

Response: The kinematic description has been added accordingly.

*Line 90: "thrusting": nw-vergent?*

Response: The top to NW kinematics are now reflected in the sentence.

*Line 93: which recumbent fold? Does it have a name? Is it visible in a figure? Any references?*

Response: Here, we refer to the recumbent fold that builds the Doldenhorn nappe. We have clarified this point and also added a reference.

*Line 95: what is an inverse succession*

Response: An inverted stratigraphic succession; now labelled accordingly.

*Line 100: I would delete "differential"*

Response: Done as suggested.

*Line 101: delete "a phase of"*

Response: done.

*Line 101: strain partitioning": only interesting to know if you describe what is partitioned into what*

Response: This has been clarified.

*Line 101: "simultaneously with dextral strike-slip": unclear. Is dextral strike slip part of the partitioning?*

Response: The strain partitioning occurred between the strike slip and NW thrusting. We have changed the structure of the sentence accordingly.

*Lines 103-104: Pfaffenkopf and Oberaar have no references*

Response: All references are given at the end of the sentence to avoid giving same references twice within one sentence.

*Lines 104-105: delete "the uplift passively rotated". The uplift is not a force… And add "were passively rotated" after "Doldenhorn Nappes".*

Response: Done.

*Line 108: which type of brittle structures?*

Response: This has been specified.

*Line 114: not necessary*

Response: We see the point and did remove the sentence (and adjusted the next one).

*Line 117-118: delete Line*

Response: We do not see why this line should be deleted since it provides information about how we have processed the data processing.

*Line 120: "only" should be before, not after mapped*

Response: Done.

*Line 121: delete only*

Response: done.

*Line 145: basement*

Response: corrected.

*Line 145: replace "are present as" with "consist of"*

Response: done

*Line 148: what is a "granodioritic texture"? Magmatic texture?*

Response: There has been a mistake in the original draft. Both coarse- and fine-grained textures are granoblastic, for which we apologize. We have corrected the sentence accordingly.

*Line 152: delete "see also"*

Response: done.

*Line 155: evidence*

Response: corrected.

*Lines 174-176: delete sentence*

Response: We agree and have deleted the sentence.

5  *Lines 180: brittle shear zone?*

Response: Changed to "brittle faults forming along former shear zones" for clarification purposes.

*Lines 181-182: "Note that…sediments" doesn't need to be between parentheses*

Response: It has to be. It is intended to point out why we start with SZ2 in the basement. Therefore, we prefer to keep it in parentheses.

10  *Line 182: replace "faulting behavoiur" with kinematics*

Response: done.

*Line 183: "However" ???*

Response: Corrected. The unfitting word "However" was removed.

*Line 183: I cannot see the offset of older structures in Fig. 5b*

15  Response: The SZ2 structures are cut in the footwall by the SZ3 thrusts running through the JSW sediments in Fig. 5b. We do find the offset rather illustrative. However, we added also a reference to Fig. 5c as this image also illustrates the cross-cutting relationship.

*Line 189: no dot after SZ2*

Response: done.

20  *Line 189: "offsets": this could give you a shear sense? Why is there no shear sense described? Looking at Fig. 6 I do not find the distinction of SZ2 and SZ3 convincing based on the orientation data. The Rottal net shows a possible distinction into two groups based on the different dip, but there aren't many data and even there it could be one population only. The Trugberg net doesn't show two"populations" in my view.*

25  Response: The offset could give us a crude information about the shear sense: top to NW for normal faulting and top to SE for reverse faulting. However, we refrain from this since we do not have lineation data for this data set (as the Trugberg set is inferred from remote sensing). For the Rottal we present information about the shear sense for SZ2 (Line 183: "dominantly top to the NW shear senses"). For SZ3 we now added a sentence where we outline the predominantly top NW-

30  directed thrusting. Regarding the distinguishing of SZ2 and SZ3 in the data for Fig. 6.: We did not differentiate the phases based on the orientation only; we rather measured 2 sets of SZ orientation in the field (Rottal) and via remote sensing (Trugberg). We thus distinguished these phases based on the combination of data of various sources. We admit that the orientation net for the Trugberg itself would not warrant a subdivision of 2 phases.

35  *Line 190: "wider": give an idea*

Response: Given in parenthesis.

*Line 190: replace "occurred" with "is"*

Response: done.

*Lines 190-193: any references to figures?*

40  Response: Reference to Fig. 5 was inserted.

*Line 193: "kinematics": you did not say much about it in the last lines (see comment on Line 189)*

Response: We acknowledge the need for more clarity. We thus changed the text accordingly. (see response to comment on line 189) and additionally added a shear sense statement. Thus, we now hope to have clearly stated what kinematics each phase is associated with.

*Lines 201-204: a verb is missing in this sentence*

Response: Corrected by removing the superficial "that"

*Line 210: I doubt it...Mapping certainly reveals something, but I guess something else.*

Response: The whole sentence is indeed unnecessary and has thus been removed.

*Line 213: why "the first"? It was not mentioned that there were many. Is it S1?*

Response: In lines 210 to 213 we describe the characteristics of S1 and label it. With this sentence we state that this is the first foliation formed in the sedimentary rocks (as it overprints the bedding but is overprinted by all later structures). We added "(S1)" after mylonitic foliation for clarification purposes

*Line 214: what is the spacing of mylonitic foliations? Spacing between shear zones?*

Response: We rephrased the sentence to clarify.

*Line 217: The reference to Fig. 7c comes after "more than one def phase", but it doesn't show that. It only shows the boudinage.*

Response: The reference was misplaced and is moved to the appropriate place in the sentence as we intend to illustrate the impressive boudins made up of dolomites. Yet it still shows minor multiphase deformation.

*Line 219: past tense (led) not always used in the text. Reference to maps and figures?*

Response: Done.

*Line 222: argument for the synchronous development of S1 and SZ1?*

Response: We do see that the statement of synchronous development is misplaced here as we give our argument for it in Sect. 5.3.1 in the discussion section. Thus, we replaced it with observations that indicate contemporary formation. (parallel orientation of S1, SZ1 and foliation spacing decreasing towards SZ1).

*Line 223: what is the shear sense of these shear zones?*

Response: This is difficult to specify due to the later deformation that partly rotated the orientation of the sedimentary stack. We do see a general top to NW trend of shearing and thrusting. We now added an according statement in a sentence to address this and to highlight the thrust nature (see next comment on Line 225).

*Line 225: it needs to be stated before that these shear zones are thrusts. And what displacement direction?*

Response: Done (see also previous comment).

*Line 225: "of" the footwall rather than "in"?*

Response: Corrected.

*Line 227: subsequent to what?*

Response: Here, we refer to the phase of thrusting (SZ1) and the contemporaneous formation of S1, which constitutes deformation stage 1 in our framework; the text was adjusted accordingly.

*Line 228: what is a rotated sigma clast? And…why does a rotated sigma clast indicate rotation of the initial structures? And… what deformation phase causes an anticlockwise rotation of some 50, looking East? "As a result": of something that is not really explained.*

Response: Misleading sentence in our previous version. Rotation is documented by rotated textures (i.e. bedding in competent units) within sigma clasts along with rotation and folding of S1 parallel veins. The rotation we discuss in this context occurs in section view looking east (not in map view). We rewrote the corresponding paragraph to clarify these points.

*Line 229: what are EN and EW?*

Response: Some regions in which the structural data was collected. This was clarified and references to Figs. 2 & 9 were added.

*Line 230: "This deformation stage": I am lost. Which one? The one that rotated the fabrics? What are the kinematics of this deformation?*

Response: The second stage rotated preexisting structures by local folding (see also response to comment on Line 231). We rewrote the corresponding paragraph for clarification purposes. We also added a kinematic description.

*Line 230: how do you know it is not preserved?*

Response: We realized that our original statement was confusing and not really needed so we removed it to increase conciseness and clarity.

*Line 231: "and the folding": so this deformation phase is the folding? What axial plane and fold axes orientation?*

Response: This deformation is characterized in the sediments by local folding near the basement cover contact, and it intensified towards the SE (= towards the more internal). Steep SE dipping axial planes and SW/W to NE/E striking folding axis are dominant, but they can be locally complex and chaotic. We now explicitly state this in the rewritten paragraph (see also responses to comments on Lines 228 & 230).

*Line 233: "Subsequent NW directed shearing": not easy to understand and visualize and I cannot really see it in Fig. 11b that is quoted at the end of the sentence. Where do I see the flat-lying limbs? What is the evidence for a subsequent NW-directed shearing in the first place?*

Response: The reference to Fig. 11b is incorrect, instead Fig. 10a is now correctly referenced. We further added several references to Fig. 7 to illustrate our points. We do see several thrusts that offset S1, S2 and SZ1 and therefore had to occur later (see. Figs. 2,10,11c). These thrusts are the reason we see basement rocks in the summit region of the Eiger-, Moench and Jungfrau summit, right on top of Mesozoic sediments. Along these thrusts the S3 foliation intensifies, and we measured a swath of stretching lineations on S3-parallel slip planes with top to NW shear sense (Fig. 9). We have adjusted the text to clarify this (see also the newly added Figs. 10a,b and the next responses to comments on Line 234).

*Line 234: "now": do you think the limbs were initially steep?*

Response: We do think that the limbs formed as intermediate-steep SE imbricate stack by fault propagation folds. The subsequent folding and thrusting led to their current flat lying position in the frontal part and folds in the internal SE part.

*Line 234: this observation of the result should be described by a photo or sketch of the structures, not by a schematic recap of the Discussion/Conclusions.*

Response: We agree and added new Figures to illustrate specifically the cross-cutting and overprinting relations of key structures (now Figs. 10a,b).

*Line 234-235: S3? How can one distinguish S3 from S2 that are sometimes parallel?*

Response: S3 formed sometimes (not always) parallel to S1 (not S2!). Often S3 foliation and SZ3 thrusts cut S1 with a small angle, yet sometimes the form at a very high angle to each other (as illustrated in Figs. 7a,b). This usually occurs when S1 structures were rotated during the phase forming S2 structures. S3 structures have a similar orientation and a shared SE to NW evolution. This evolution is characterized by SE dipping in the SE to flat and slightly NW plunging in the NW (Figs. 10,11). We adjusted the text of the whole section to better describe these observations. We also added the new Figures (10b,c) to further document the cross-cutting and the incorporation of basement slivers at different stages.

*Line 236: difficult to follow. Why is it really necessary to distinguish an S3 from an S2 here?*

Response: We actually distinguish S3 from S1. S1 is folded by stage 2 deformation that produced a weak S2 axial plane foliation. Both are cut by S3 structures.

*Line 237: which one is the "same" orientation?*

Response: We intend to mention that S3 features a fairly 'consistent' orientation throughout the study area; see response to comments on Lines 234-235.

*Lines 210-240: not easy to follow, especially the mixture of orientation with cross-cutting relationships and of microstructural characteristics.*

Response: We saw the need to rephrase the entire paragraph for clarification purposes.

*Line 243: what does oblique mean here? Kinematics of these faults?*

Response: Oblique refers to the general orientation of the faults as well as the shear sense (strike slip with reverse/normal fault behavior). We amended the text to reflect that.

*Line 258: specify and justify the geothermal gradient used for the conversion of T in depth*

Response: We use a uniform averaged geothermal gradient of 27 °C km$^{-1}$ for the upper crust as often given with a range of 25° to 30° C km$^{-1}$ for the continental lithospheric crust (Pollack and Chapman, 1977). It seems reasonable for the youngest exposure history of the Aar massif (~26-28 °C km$^{-1}$; Valla et al., 2016; Schlunegger and Willett, 1999) and is close to the inference by Glotzbach et al. (2010) of 25 °C km$^{-1}$. The reason for the depth estimation was to highlight the rather shallow position during peak T of Alpine deformation (especially compared to southern Aar massif; Herwegh et al., 2017). We recalculated the depth estimation for the given gradient of 27° C km$^{-1}$ and added statements in in the main text to address this properly with 3 new references.

*Line 266: evidence for being deformed as an "ensemble"?*

Response: They share the same deformation fabric, same RSCM peak temperature, which we describe in Sect. 4.2.2 (as it is now referenced in the sentence). We have clarified this point.

*Line 267: consistent? With what?*

Response: Internally consistent. Sentence was changed accordingly.

*Line 268: less shortening: evidence? Quantified?*

Response: We did not (yet) quantify the individual offset along strike, but gave a qualitative statement based on the map view. We added a statement to clarify this point.

*Lines 269-270: why?*

Response: The steepening occurred mostly during SZ2, due to the large differential block uplift in the basement (SZ2) that passively steepened the sedimentary cover in the SE. A statement was added to clarify this.

*Line 272: The cover-sediment interface?*

Response: We wrote "cover sediments" which we rephrased to "sedimentary cover" to avoid possible confusions.

*Line 283: why "already"?*

Response: No need for this; deleted.

*Line 284: has this wedging been described before in the text? Wedging associated with folding? I am lost…*

Response: This particular wedging-in refers to a pre-Alpine event, which we take from literature survey. An additional reference was added, where an up-to-date and extensive discussion of this Pre-Alpine evolution is presented. We want to highlight the already existing structures before the Mesozoic.

*Line 286: evolution of what?*

Response: The pre-Alpine basement gneisses. Sentence was amended accordingly.

*Line 289: succeeding???*

Response: changed to Alpine.

*Line 293: favorable for what?*

Response: for localization of Jurassic normal faults. Sentence was adjusted to clarify.

*Lines 295-301: I am not sure why these observations are important in the context of the results session and of the paper.*

Response: We do generally find the relative consistency of the strata remarkable. We briefly discuss the main trends in the stratigraphy that allows us to differentiate tectonic from stratigraphic features of the map (Fig. 2). We consider this brief section essential for our understanding of how we can differentiate and map the different tectonic slivers.

*Lines 302-305: why to mention an own stratigraphic model here that was not presented in the results?*

Response: We do present a stratigraphic model in the Appendix A and discuss the main findings in Sect. 4.1.2. The reason for giving the details in the Appendix is that for almost the entire stratigraphic units we confirm previous field data, thus there is not much new insight. Yet most references are quite old, and literature spans several decades, leading often to different and outdated interpretations. Furthermore, there is no consistent and up-to-date stratigraphic model for the region. We thus saw the need to conduct such a compilation (see also previous 2 comments).

*Lines 305-307: I am still not sure about the importance of these lines.*

Response: See previous 3 comments and responses, respectively.

*Line 311: Substratum T constrained by RSCM?*

Response: Since the sedimentary cover is still in place on top of the substratum, the inferred Alpine RSCM temperatures are also constraining the T in the underlying basement. We added "Alpine" to the sentence to clarify that we explicitly address the shared peak T.

*Line 311: why to give an upper limit to calcite thermometry and mention calcite thermometry at all?*

Response: The sentence was misleading. It should convey that the upper temperature constraint is given by the RSCM at around ~320 °C. We rephrased the sentence to clearly state that now.

*Line 314: ok, but you should also quote the lower T suggested by XX and White. I think it must be Burkhard, 1993, not Burkhard, 1990.*

Response: Done. We adapted the lower boundary to the lower boundary suggested by Kennedy & White (2001) and reference the work. Burkhard reference was corrected.

*Line 322-323: repetition of line 317*

Response: Very similar indeed. To avoid redundancy and increase conciseness we removed the last 2 sentences of the paragraph (see also comment and response below).

*Line 323: well, I am not sure that this can be constrained. The only thing that can be said is that T of 270°C was attained following the recrystallization T inferred by Stipp et al., 2001. What is known about the T peak anyway?*

Response: Indeed, we can only constrain the formation to the T window of <330 °C to 270 °C. Since from our data we do not have chronological constraints, we cannot state how close (or not) to peak T the deformation occurred. Therefore, we removed the last sentence.

*Line 325: pelitic*

Response: Corrected.

*Line 325: synchronously: what is the evidence?*

Response: We see a parallel orientation of S1 and SZ1; both share the same response to the later deformation (i.e. being cut or modulated by local folding). The evidence for the synchronous formation is now presented with more clarity in Sect. 4.2. and Figs. 10a,b.

*Line 326: delete "rheological"*

Response: done.

*Line 333: was it quantified? What is the evidence?*

Response: It was not quantified by retro-deformation; the presented several-km shortening is an estimation and is derived from the preserved offsets within the map (Fig. 2) and the profile (Fig. 10). The thrusts themselves are easily identifiable (Collet & Parejas, 1931) in the field (i.e. by the offset of the Mesozoic strata and basement rock incorporation; see also Figs. 2,10). The evidence for the thrusts is now presented with more clarity in Sect. 4.2. and Figs. 10a,b.

*Line 344: was really shown that incorporation of basement slivers in the cover is associated to the 1$^{st}$ deformation phase?*

Response: We do agree there is need to better show this wedging-in. We do so by adding Figs. 10a,b to explicitly illustrate the incorporation and the resulting cross-cutting relations.

*Line 361: how significant?*

Response: At least > 2 km in the JSW in the Rottal section (as can be seen in Figs. 2 & 10). An related statement was added in parenthesis with reference to Fig. 10.

*Line 365: repetition of 361*

Response: Removed to avoid redundancy.

*Line 369: so why should they be called shear zones?*

Response: We address this by clarifying the wording and now calling them faults (F1 for the steep faults; "Gadmen" of Berger et al. 2017; and F2 for the oblique to strike slip set; "a-c" joints of Ustazeswksi et al., 2007). We rephrased the last paragraph of Sects. 4.2.1, 4.2.2 and concerning parts in section 5.3.3. We corrected the names in Fig. 9 and added a clarification to the caption of Fig. A2.

*Line 380-384: So S1 is at < 330_C, shows dynamic rexx of quartz, but no fabric in the basement, as stated in Line ...?*

Response: We do indeed not see any related deformation in the basement, otherwise we would expect to find some fabric formed by dynamic recrystallization. S1 and SZ1 are only found in the sedimentary cover.

*Lines 386 – 388: so … under which T conditions did these shear zones develop?*

Response: The structures of SZ1 or SZ2 formed under similar conditions, maybe with slightly lower temperatures for SZ2. We stated so previously in Sect. 4.2 and do now so in more clarity in the revised manuscript (see also comments and responses on Lines 183 to 266).

*Lines 390-392: An amazing change of scale of interpretation!*

Response: We rephrased the sentence to clearly indicate the large-scale linkage to orogenic processes.

*Lines 394-395: 3remaining compressional orogenic forces": I think these speculations are not really necessary.*

Response: We agree that this discussion goes beyond the scope of this paper, and therefore (and to ensure a concise paper) the half-sentence was removed.

*Line 400: "aggravated": aggravating a link? First, it sounds quite dramatic, second it's not the link, but its interpretation that may be "aggravated"*

Response: Corrected to "characterized".

*Line 404: "is key": structure of the sentence needs to be reconstructed.*

Response: The sentence was split into two and the redundant "is key" was deleted.

*Line 405: "and discrete":?*

Response: Unnecessary words were removed to make the sentence more concise.

*Lines 404-406: "while" and "whereas" is too much for the same sentence. Needs to be reformulated.*

Response: "whereas" changed to "despite".

*Line 406: "bulk of the rock behaved in a brittle manner": do you mean that between the shear zone the rock was also deforming but in a brittle manner? Not clear.*

Response: Clarified by specifically labelling the bulk of the crystalline basement rocks as the brittle behaving.

*Line 410: "multiphase tectonics": strange term.*

Response: Changed to "multiphase deformation".

*Lines 411- 412: "the structural imprint … .sets up the stage for erosion": strange statement. I guess the authors wish to say that the steeply oriented displacements created uplift and exhumation by erosion?*

Response: This is a misunderstanding. In fact, the steep displacements in combination with the subsequent thrusts provided ideal boundary conditions for preferential erosion to produce the morphological contrast in front of the Aar massif. The sentence was modified accordingly.

**References**

[revised manuscript text omitted]